# Design of a two-dimensional interplanar heterojunction for catalytic cancer therapy

Yong Kang [1,4], Zhuo Mao[1,4], Ying Wang[2], Chao Pan[1], Meitong Ou[2], Hanjie Zhang[3], Weiwei Zeng[3] & Xiaoyuan Ji[1✉]

Limited substrates content is a major hurdle dampening the antitumor effect of catalytic therapy. Herein, a two-dimensional interplanar heterojunction (FeOCl/FeOOH NSs) with ·OH generation under ultrasound irradiation is fabricated and utilized for catalytic cancer therapy. This interplanar heterojunction is prepared through replacing chlorine from iron oxychloride with hydroxyl. Benefiting from the longer hydroxyl bond length and enhanced affinity with water, the alkali replacement treatment integrates interplanar heterojunction synthesis and exfoliation in one step. In particular, a build-in electric field facilitated Z-scheme interplanar heterojunction is formed due to the aligning Fermi levels. The holes on the valence band of FeOCl have great ability to catalyze $O_2$ evolution from $H_2O$, meanwhile, the generated $O_2$ is immediately and directly reduced to $H_2O_2$ by the electrons on the conductive band of FeOOH. The self-supplying $H_2O_2$ ability guarantees efficient ·OH generation via the Fenton-like reaction catalyzed by FeOCl/FeOOH NSs, which exhibits excellent anti-tumor performance.

[1] Academy of Medical Engineering and Translational Medicine, Medical College, Tianjin University, Tianjin 300072, China. [2] School of Pharmaceutical Sciences (Shenzhen), Sun Yat-sen University, Guangzhou 510275, China. [3] Tianjin Key Laboratory of Biomedical Materials, Key Laboratory of Biomaterials and Nanotechnology for Cancer Immunotherapy, Institute of Biomedical Engineering, Chinese Academy of Medical Sciences and Peking Union Medical College, Tianjin 300192, China. [4]These authors contributed equally: Yong Kang, Zhuo Mao. ✉email: jixiaoyuan@tju.edu.cn

Chemodynamic therapy (CDT), a representative catalytic therapy, is a burgeoning treatment strategy that applies Fenton or Fenton-like reactions to generate hydroxyl radical (·OH) with high cytotoxicity for inducing cancer cell apoptosis[1–4]. The production of ·OH via catalyzing the disproportionation reaction of hydrogen peroxide ($H_2O_2$) depends on Fenton chemistry, which requires neither external energy input nor oxygen ($O_2$) as substrate[5–7]. Such a distinct mode of reactive oxygen species (ROS) production enables CDT to circumvent the primary obstacles of tumor hypoxia-associated resistance and limited light penetration depth in cancer photodynamic therapy (PDT)[8,9]. Accordingly, much research effort has been dedicated to the development of CDT. In the past decades, various reductive ions, such as Fe-based and other metal-based nanomaterials with the capability of catalyzing Fenton-type reactions have been developed for tumor CDT by using endogenously produced $H_2O_2$ as the substrate[3,10–18]. However, although the intracellular $H_2O_2$ level in most cancer cells is higher than that in normal cells, the endogenous $H_2O_2$ is still not nearly enough to meet the demand for satisfactory chemodynamic efficacy[10,19–22]. As far as this is concerned, the introduction of $H_2O_2$ self-supplying ability into CDT agents is highly desirable for achieving a satisfactory anticancer effect.

Two different patterns have been developed to improve the content of intracellular $H_2O_2$: enhancement of endogenously mitochondrial production of $H_2O_2$, or supplement via exogenous $H_2O_2$ generation or delivery[10,17,23]. Nevertheless, the endogenous $H_2O_2$ produced in mitochondria will inevitably be decomposed rapidly by cellular scavenging enzymes such as peroxidases and catalase before approaching CDT agents to initiate Fenton-type reactions, leading to poor practicability of endogenously produced $H_2O_2$ for enhancing chemodynamic efficacy[10]. In addition, the delivery of $H_2O_2$ or its precursor by nanocarriers to enhance intracellular $H_2O_2$ concentration was limited by low delivery capacity, random release, and unsustainability[24]. Currently, the most used $H_2O_2$ producers, glucose oxidase and its mimics, collaborated with a catalyst to yield ·OH, which enhances chemodynamic efficacy to some extent[20,25,26]. However, the glucose oxidase and its mimics mediated $H_2O_2$ production from glucose and $O_2$ are severely limited by the undesired elimination during blood circulation and tumor hypoxia[27–29]. Despite enormous struggles, engineering CDT agents with specific and efficient $H_2O_2$-supplementing functionality without being limited to the tumor microenvironment (TME) remains a great challenge.

In our previous studies, two-dimensional (2D) heterojunction with independent and separated oxidation and reduction centers have revealed great potential in photo-catalyzed water splitting, $CO_2$ reduction, and photo-induced cancer catalytic therapy[30–34]. Specifically, the separated electrons in the conduction band (CB) of one catalyst with high reduction potential are able to catalyze various reduction reactions, such as $O_2/·O_2^-$, $CO_2/CO$, $O_2/H_2O_2$, et al., meanwhile, the separated holes in the valence band (VB) of the other catalyst with high oxidation potential also have ability to mediated diverse oxidation reactions, such as GSH/GSSG, $OH^-/·OH$, $H_2O/O_2$, et al. Therefore, 2D heterojunction with ability to realize reduction and oxidation reactions synchronously provides an alternative detour strategy for some reactions, which are difficult to be realized in one step, for example, in situ generation of $H_2O_2$ through two-step circuitous strategy ($H_2O \xrightarrow{h+} O_2 \xrightarrow{e-} H_2O_2$) to break through the limitations of the TME.

High expression of $H_2O_2$ is one of the unique microenvironments of tumor cells and a specific target of chemodynamic tumor therapy. However, the endogenous $H_2O_2$ is still not nearly enough to meet the demand for satisfactory chemodynamic efficacy[19–22]. Moreover, the reported $H_2O_2$ supplying strategies to

improve chemodynamic efficacy were restricted to safety, targeting, and tumor hypoxic microenvironment. Hence, engineering CDT agents with specific and efficient $H_2O_2$-supplying functionality without being limited to the TME is a huge breakthrough for clinic development of CDT and cancer treatment.

In this work, we develop a smart wet-chemical strategy based on an alkali replacement reaction that can intelligently integrate interplanar heterojunction synthesis and two-dimensional ultrathin heterojunction exfoliation in one step. As shown in Fig. 1, the bulk layered iron oxychloride (FeOCl) is synthesized by the hydrothermal process using iron chloride ($FeCl_3$) as a substrate. Afterward, the alkali etching strategy is applied to replace chlorine (–Cl) with hydroxyl (–OH), which not only widens the interlayer space but also increases interlayer water molecule infiltration due to the longer hydroxyl bond length and enhanced affinity with water molecules. Under probe sonication, 2D ultrathin heterojunction (FeOCl/FeOOH) with few nanometers are easily exfoliated. After PEGylation with PEG-$NH_2$, the engineering FeOCl/FeOOH nanosheets (NSs) based interplanar heterojunction with efficient $H_2O_2$ self-supplying ability using $H_2O$ as the only substrate is developed and exhibits a highly efficient chemodynamic effect. In the interplanar heterojunction FeOCl/FeOOH NSs, FeOCl part and FeOOH part with different Fermi levels and energy band structures contacting with each other induces charges to redistribute at their interfaces due to the aligning Fermi levels, which mediates the construction of a built-in electric field in their interface. Upon US irradiation, the US-excited electrons on the CB of FeOCl is combined with the holes on the VB of FeOOH guided by the built-in electric field in their interface, leaving stronger reduction/oxidation potentials of separated electrons and holes on the CB of FeOOH and the VB of FeOCl. Meanwhile, a Schottky barrier is formed owing to the band bending at their interface, which further suppresses the electron flow from FeOOH to FeOCl and enhances their Z-schemed charges transfer. Hence, a built-in electric field and Schottky barrier facilitated Z-schemed catalytic mechanism is constructed, in which the holes on the VB of FeOCl have great ability to oxidize $H_2O$ and produce $O_2$, meanwhile, the generated $O_2$ is immediately and directly reduced to $H_2O_2$ by the electrons on the CB of FeOOH. The self-supplying $H_2O_2$ guarantees the continuously and endlessly ·OH generation via the Fenton-like reaction catalyzed by both FeOCl and FeOOH. The obtained FeOCl/FeOOH NSs with highly efficient chemodynamic effect exhibit excellent anti-tumor performance both in vitro and in vivo. Hence, this research not only affords a smart method to synthesize two-dimensional ultrathin heterojunction, but also demonstrates a proof-of-concept application of the obtained 2D heterojunction in tumor catalytic therapy, which may serve as a modest spur to induce future valuable contributions to other possible fields.

## Results

**Preparation and characterization of FeOCl/FeOOH NSs-based interplanar heterojunction.** The FeOCl/FeOOH NSs-based interplanar heterojunction was synthesized through a wet-chemical strategy based on an alkali replacement reaction. Detailly, as illustrated in Fig. 1a, the bulk layered FeOCl with a color of the wine was synthesized by the hydrothermal process using $FeCl_3$ as substrate (Supplementary Fig. 1). As shown in Fig. 2a, the size of the FeOCl powder was about 500 nm, and the layered structure was clearly exhibited in the scanning electron microscope (SEM) image of FeOCl (Fig. 2a, b). In addition, the energy dispersive spectrometer (EDS) mappings observed in Fig. 2i exhibited that these characteristic elements Fe, O, and Cl were uniformly distributed in FeOCl NSs, demonstrating the successful fabrication of FeOCl. In order to compare with FeOCl/FeOOH NSs, the FeOCl NSs were firstly prepared through liquid

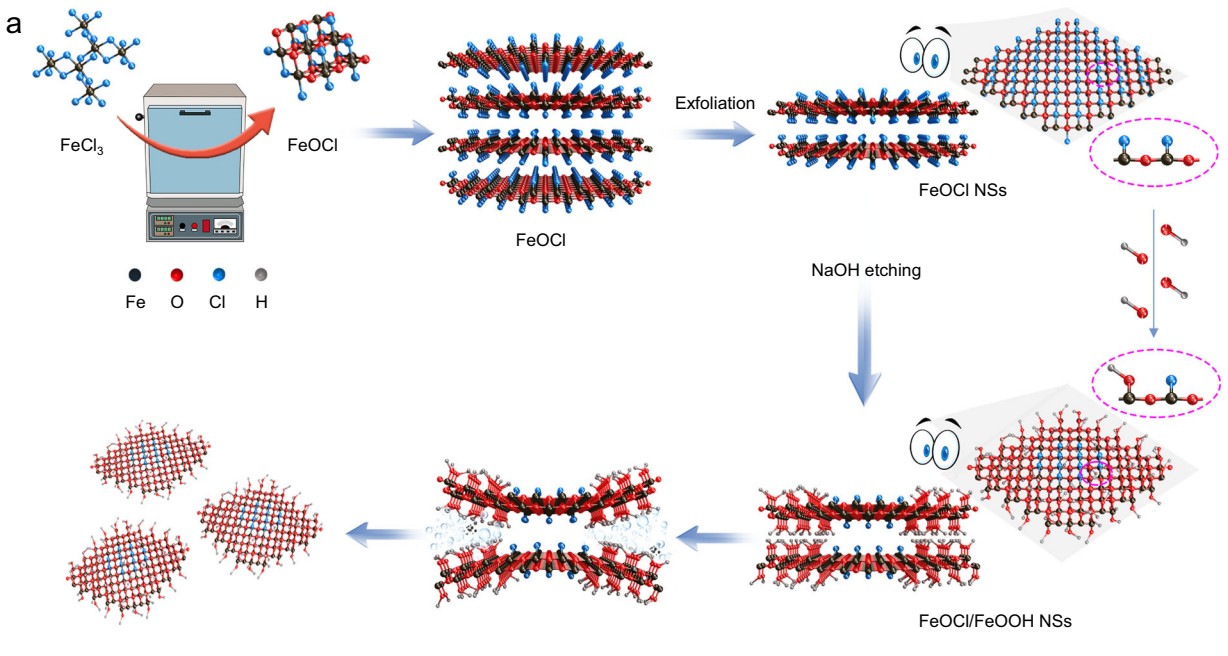

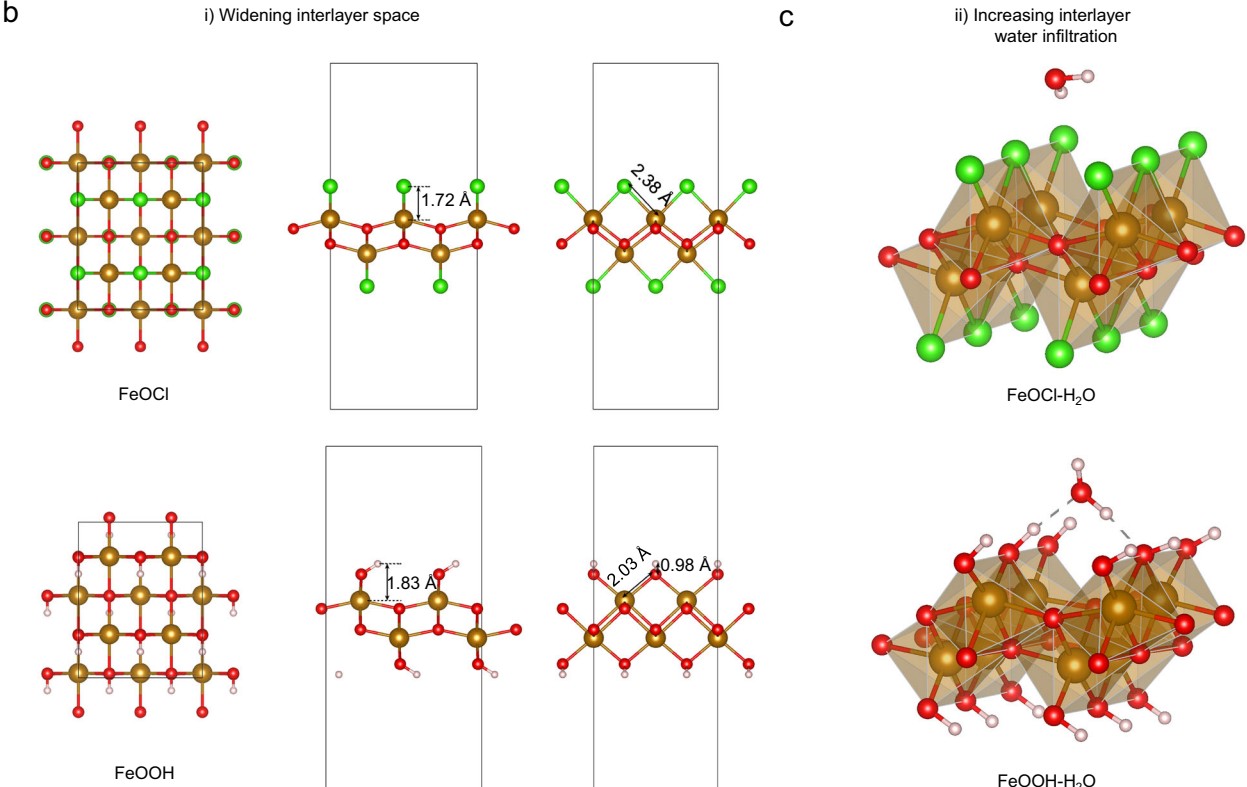

**Fig. 1 Schematic illustration of preparation of FeOCl/FeOOH nanosheets (NSs) for efficient catalytic cancer therapy. a** FeOCl/FeOOH NSs with Z-schemed heterojunction were fabricated by hydrothermal process, liquid exfoliation, and NaOH etching. A smart wet-chemical method based on alkali replacement reaction was developed to intelligently integrate interplanar heterojunction synthesis and two-dimensional ultrathin heterojunction exfoliation in one step. **b** The fully optimized structures of the FeOCl and FeOOH NSs with caculated bond length of Fe-Cl in FeOCl and bond lengths of Fe-O and O-H in FeOOH. **c** The most stable adsorption configurations of $H_2O$ molecule on FeOCl and FeOOH NSs.

exfoliation of layered FeOCl powder in water via high-intensitive probe sonication. After 12 h of continuous sonication, the FeOCl NSs with the size of 120 nm were fabricated (Supplementary Fig. 2a). The transmission electron microscopy (TEM) images of FeOCl NSs were observed in Fig. 2c, and the high-resolution

transmission electron microscopy (HRTEM) exhibited a distinct interference fringe and *d*-spacing of 0.395 nm corresponding to the plane of FeOCl (Fig. 2d), which provided direct evidence for the successful fabrication of the FeOCl NSs. For the preparation of FeOCl/FeOOH NSs-based interplanar heterojunction, the

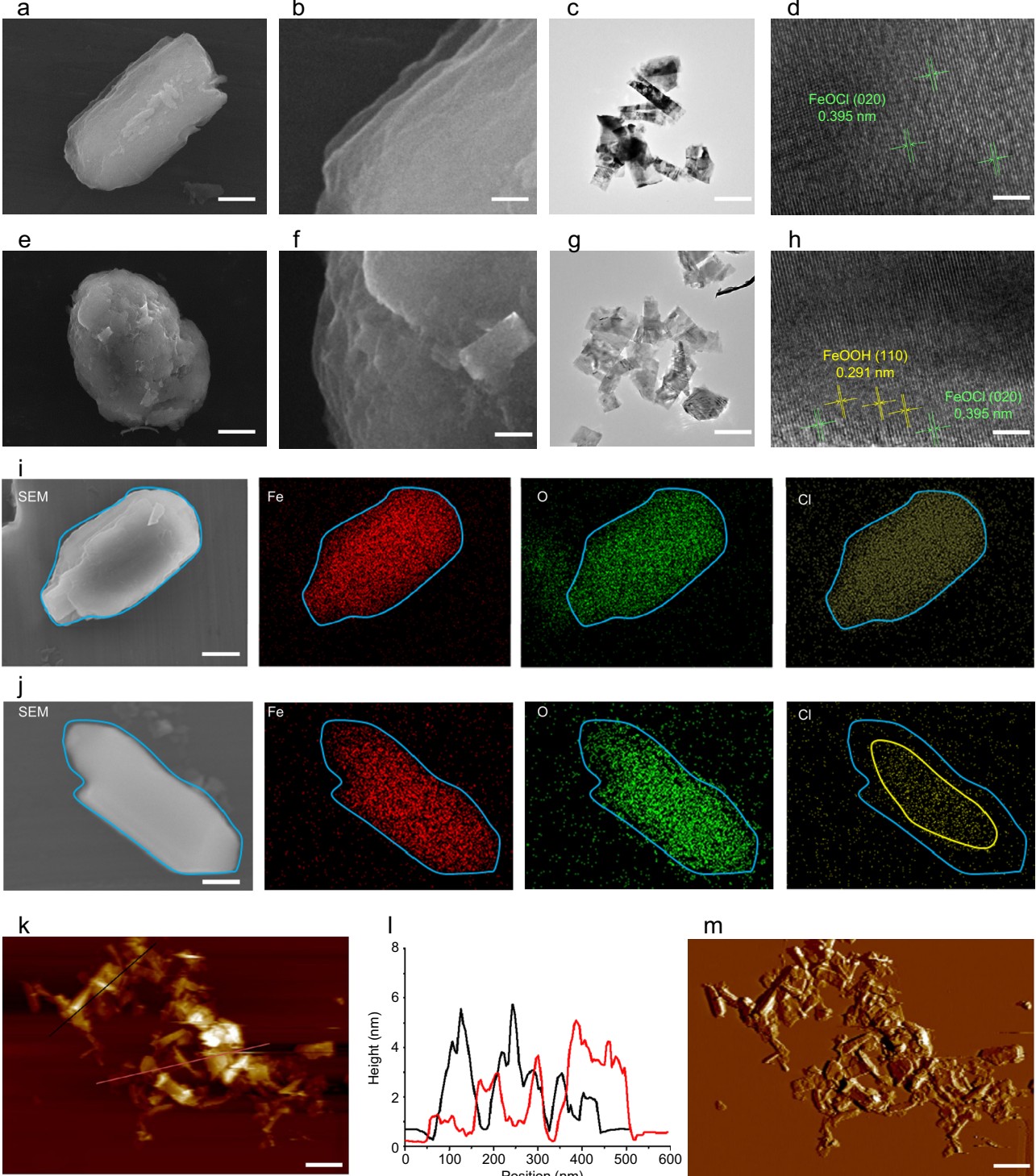

**Fig. 2 Morphology and composition characterization of layered FeOCl NSs and FeOCl/FeOOH NSs.** SEM images of layered **a** FeOCl and **e** FeOCl/FeOOH, scale bar = 100 nm. Enlarged SEM images of layered **b** FeOCl and **f** FeOCl/FeOOH, scale bar = 15 nm. TEM images of ultrathin **c** FeOCl NSs and **g** FeOCl/FeOOH NSs, scale bar = 100 nm. HRTEM images of ultrathin **d** FeOCl NSs and **h** FeOCl/FeOOH NSs, scale bar = 3 nm. SEM-EDS mapping of **i** FeOCl and **j** FeOCl/FeOOH: red (Fe), green (O), and yellow (Cl), scale bar = 100 nm. **k** AFM images of FeOCl/FeOOH NSs, scale bar = 100 nm. **l** Thickness of FeOCl/FeOOH NSs. **m** 3D AFM images of FeOCl/FeOOH NSs, scale bar = 100 nm. For these morphology characterizations of fabricated FeOCl NSs and FeOCl/FeOOH NSs, three times each experiment was repeated independently with similar results. The units (arb. units) of intensities of XPS, XRD, and Raman spectra mean arbitrary units. Source data are provided as a Source Data file.

alkali etching strategy was applied to replace chlorine with hydroxyl. As shown in Supplementary Fig. 1, the color of FeOCl/FeOOH NSs became orange from the color of wine after alkali etching treatment. Figure 2e, f revealed the morphology of bulk FeOCl/FeOOH, in which the layered structure and a certain degree of curling around the edges of each layer were observed. The EDS mapping provided direct evidence for the replacement of chlorine with hydroxyl during alkali etching (Fig. 2j), in which Fe and O were still homogeneously distributed in whole NSs. However, Cl focused only on the central area of NSs. The replacement of chlorine with hydroxyl during alkali etching might be the main reason for the changes in their morphology. Afterward, a mild probe sonication was applied to exfoliate ultrathin FeOCl/FeOOH NSs from their bulk. After only 4 h of probe sonication at a mild intensity, the ultrathin FeOCl/FeOOH NSs with the size of about 115 nm were obtained (Supplementary Fig. 2b). Although the conditions and time for exfoliation of FeOCl/FeOOH NSs were much milder and shorter than that of FeOCl NSs, the TEM images of ultrathin FeOCl/FeOOH NSs, exhibited in Fig. 2g, were much thinner compared with the TEM images of FeOCl NSs from the contrast point of colors. The ultrathin layer of FeOCl/FeOOH NSs also was confirmed by the atomic force microscopy (AFM) images, in which the thickness of FeOCl/FeOOH NSs was decreased to about 3 nm (Fig. 2k–m). We attribute this easier-to-peel nature of FeOCl/FeOOH NSs to the alkali replacement reaction, which not only widens the interlayer space but also increases interlayer water molecule infiltration due to the longer hydroxyl bond length and enhanced affinity with water molecules. In order to deeply dig the effect of alkali replacement on the synthesis of this FeOCl/FeOOH NSs-based interplanar heterojunction, spin-polarized density functional theory (DFT) calculations were applied. First, the fully optimized structures of the FeOCl and FeOOH NSs are shown in Fig. 1b. The bond length of Fe-Cl in FeOCl is measured to be 1.72 Å. For FeOOH, the measured bond lengths of Fe–O and O–H are 2.03 and 0.98 Å, respectively. And the bond angle of Fe–O–H is measured to be 111.5°. The calculation results show that the distance between Fe and the outermost atoms (Cl in FeOCl or H in FeOOH) is elongated from 1.72 to 1.83 Å after –Cl was substituted to –OH. Second, to compare the hydrophilic property of FeOCl and FeOOH, the adsorption characteristics of these two substrates towards $H_2O$ molecule are investigated. The most stable adsorption configurations of $H_2O$ molecule on FeOCl and FeOOH nanosheets are demonstrated in Fig. 1c. Supplementary Table 1 contains the equilibrium distance, the intramolecular bond angle and bond length of $H_2O$ before and after adsorption, the adsorption energy, and charge transfer of each system. The calculated results show that $H_2O$ molecule binds on the surface of FeOOH with a closer distance than that in FeOCl. Adsorption behavior also leads to significant changes in the bond angle of $H_2O$ molecule, which decreased and increased by 1.779° and 0.704° in FeOCl and FeOOH systems, respectively. Note that in FeOOH adsorption system, the bond of O–H in $H_2O$ close to the surface is elongated from 0.972 to 1.101 Å, which makes a hydrogen bond of 1.25 Å formed. The other hydrogen bond with a length of 2.24 Å is contributed by the O in $H_2O$ and the H in FeOOH. Further, the adsorption energies of FeOCl and FeOOH towards $H_2O$ molecule are calculated to be −0.13 and −0.83 eV, respectively (Supplementary Table 2). The corresponding charge transfer is 0.001 and −0.006 respectively (Supplementary Table 3). The small absolute terms of the adsorption energy and the negligible charge transfer indicate that although the adsorption process could proceed spontaneously, the interaction between FeOCl nanosheet and $H_2O$ is very weak. While, the obtained absolute term of adsorption energy is remarkably higher, indicating the stable binding between $H_2O$ and FeOOH.

These findings support that FeOOH has stronger hydrophilic properties than FeOCl. The mechanism can be explained as that the sites that form intermolecular hydrogen bonds with $H_2O$ increase after −Cl was substituted to −OH, which greatly improves the hydrophilic property. In addition, the HRTEM images showed the distinct interference fringe and $d$-spacing of 0.395 nm and 0.291 nm, which corresponded to the plane of FeOCl NSs and FeOOH NSs (Fig. 2h).

Next, X-ray diffractometry (XRD), X-ray photoelectron spectroscopy (XPS), and Raman spectra were performed to analyze the chemical composition and structures of as-prepared FeOCl NSs and FeOCl/FeOOH NSs. In the XPS spectra (Fig. 3a), the specific peaks of Fe $2p$, O $1s$, and Cl $2p$ were observed, respectively. The typical high-resolution XPS spectra of Fe $2p$ of FeOCl NSs was exhibited in Fig. 3d, the spectra were made up of four peaks at 711.75, 725.45, 718.25, and 731.85 eV, which were attributed to $Fe^{3+}$ $2p_{3/2}$, $Fe^{3+}$ $2p_{1/2}$, $Fe^{3+}$ $2p_{3/2}$ satellite, and $Fe^{3+}$ $2p_{1/2}$ satellite, respectively. For the typical high-resolution XPS spectra of Fe $2p$ of FeOCl/FeOOH NSs, except for these specific peaks of Fe $2p$ FeOCl NSs, another two specific peaks of Fe $2p$ FeOOH NSs were also observed, including $Fe^{3+}$ $2p_{3/2}$ and $Fe^{3+}$ $2p_{1/2}$ at 712.80 and 726.60 eV, further demonstrated the successful fabrication of FeOCl/FeOOH NSs heterojunction (Fig. 3g). More obvious evidence was exhibited in the high-resolution XPS spectra of O $1s$ and Cl $2p$. As shown in Fig. 3e, h, the replacement of chlorine with hydroxyl via alkali etching strategy was observed, in which the intensity of Cl $2p$ in FeOCl/FeOOH NSs dropped dramatically compared to that in FeOCl NSs. Additionally, the peak height and width of O $1s$ in FeOCl/FeOOH NSs were increased, which ascribed that the introduced hydroxyl group replaced Cl (Fig. 3f, i). In the XRD spectra of FeOCl NSs (Fig. 3b), all the XRD peaks are well-matched with JCPDS card No. 01-0081 corresponding to orthorhombic structured FeOCl nanocrystals, which demonstrated the high purity of synthesized FeOCl. After alkali etching, another respective crystal structures were observed, which corresponded with FeOOH (JCPDS No. 01-0136), illustrating the successful edge decoration. In the Raman spectra (Fig. 3c), the peaks of FeOCl at $A_{1g}$ mode (212 cm$^{-1}$) and Eg mode (291 cm$^{-1}$) are related to Fe−O stretching vibration. After alkali etching, the characteristic peaks of FeOOH at 240 cm$^{-1}$ and 400 cm$^{-1}$ were all exhibited in the spectrum of FeOCl/FeOOH NSs, which further demonstrated the successful edge decoration. Due to the biocompatibility and dispersibility are crucial for biomedical applications, the as-prepared FeOCl NSs and FeOCl/FeOOH NSs were modified by PEG-NH$_2$. PEG(5k)-NH$_2$ positively charged was absorbed on the negatively charged surface of prepared FeOCl NSs and FeOCl/FeOOH NSs (Supplementary Fig. 3) via electrostatic attraction. The UV-vis-NIR absorbance spectra exhibited that there was no significant difference before and after PEG modification (Supplementary Fig. 4). The amount of PEG(5k)-NH$_2$ that was coated on the surface of the FeOCl/FeOOH NSs was ≈22.4% (w/w) as measured by thermo gravimetric analysis (TGA). PEGylation of FeOCl NSs and FeOCl/FeOOH NSs possess an enhanced dispersion in water, phosphate buffer saline (PBS), and cell culture medium compared with bare NSs (Supplementary Fig. 5). So that, more catalytic active sites of PEGylated nanocatalysts were exposed to substrates than that of aggregated nanocatalysts. In addition, since the catalytic substrates are dissolved in water, PEG modification can effectively increase the surface hydrophilicity of nanomaterials, and thus increase the adsorption of substrate molecules, such as dissolved $O_2$, GSH, and $H_2O$, on the surface of prepared NSs. Therefore, PEGylation of prepared NSs or other nanocatalysts is necessary for safeguarding their catalytic activity. More important, for intravenously injected nanomedicines, PEG modification is an important means to ensure that nanomedicines are not cleared by immune cells, so as to fully guarantee the

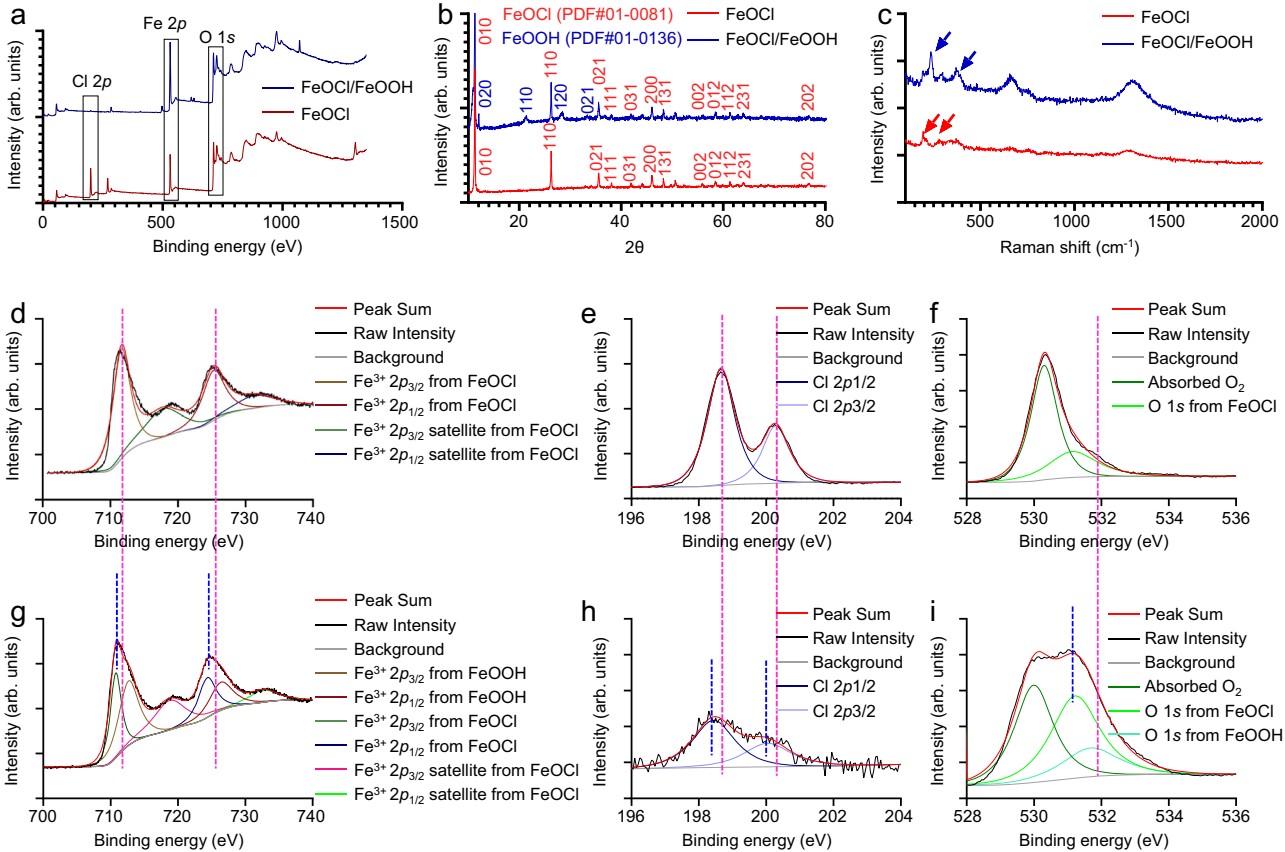

**Fig. 3 Chemical composition and structure characterization of layered FeOCl NSs and FeOCl/FeOOH NSs. a** XPS spectra of FeOCl NSs and FeOCl/FeOOH NSs. **b** XRD spectra of FeOCl NSs and FeOCl/FeOOH NSs. **c** Raman shift spectra of FeOCl NSs and FeOCl/FeOOH NSs. **d, g** HRXPS spectra of Fe 2p in FeOCl NSs and FeOCl/FeOOH NSs. **e, h** HRXPS spectra of Cl 2p in FeOCl NSs and FeOCl/FeOOH NSs. **f, i** HRXPS spectra of O 1s in FeOCl NSs and FeOCl/FeOOH NSs. For these characterizations of fabricated FeOCl NSs and FeOCl/FeOOH NSs, including XPS, XRD, Raman shift, and HRXPS, three times each experiment was repeated independently with similar results. The units (arb. units) of intensities of XPS, XRD, and Raman spectra mean arbitrary units. Source data are provided as a Source Data file.

circulation time and tumor enrichment of nanomedicines in vivo. Therefore, PEGylation is essential for nanomedicine used in vivo, and all of the catalytic properties, in vitro, and in vivo experiments in this research were carried out after PEGylation of FeOCl NSs and FeOCl/FeOOH NSs. In addition, the Fourier transform infrared (FT-IR) absorption spectra exhibited these specific bands of C=O and –CH at ~1250 cm$^{-1}$ and ~2900 cm$^{-1}$, further demonstrated the modification of PEG (Supplementary Fig. 6).

**Analysis of catalytic performance and mechanism.** As our previous reports, some metallic compounds, especially Fe-based compounds, showed great potential for mediating Fenton-type reactions[3,31,33]. In this study, methylene blue (MB) was selected as an indicator to detect the ability of ·OH generation of FeOCl NSs and FeOCl/FeOOH NSs via Fenton reactions. Figure 4a, b and Supplementary Fig. 7 displayed the degradation of MB within 30 min under different treatments with 50 μM $H_2O_2$ as substrate. For the FeOCl NSs group, there was a gentle decrease in MB content, meaning FeOCl NSs could catalyze the disproportionation reaction of $H_2O_2$ to generate ·OH. Comparably, a slightly enhanced degradation of MB was observed treated with FeOCl/FeOOH NSs and $H_2O_2$, which might attribute to the ultrathin layered structure of FeOCl/FeOOH NSs exposing more catalyzing sites compared with FeOCl NSs. Interestingly, a much obvious and faster degradation of MB was exhibited coupling FeOCl NSs with US irradiation, which should be ascribed to the US treatment

promotes ionization of $Fe^{2+}/Fe^{3+}$ and enhances the ·OH generation. Moreover, the fastest degradation of MB was observed in FeOCl/FeOOH NSs + US group. Similar to FeOCl NSs + US group, the promoting ionization of $Fe^{2+}/Fe^{3+}$ under US irradiation should be one of the reasons for the efficient ·OH generation. We speculated that the improved concentration of $H_2O_2$ might be the prime reason for such a great extent improvement of ·OH generation. In order to verify our speculation, the degradation of MB under FeOCl NSs + US and FeOCl/FeOOH NSs + US treatments without extra added $H_2O_2$ were tested. For FeOCl NSs + US without $H_2O_2$ group, there was negligible degradation of MB, demonstrating the source of ·OH is the extra added $H_2O_2$. Surprisingly, a very obvious and fast ·OH generation was exhibited under FeOCl/FeOOH NSs + US treatment even without any extra added $H_2O_2$. This interesting phenomenon demonstrated the $H_2O_2$ self-supplying ability of FeOCl/FeOOH NSs coupling with US irradiation. To further confirm the $H_2O_2$ self-supplying ability of FeOCl/FeOOH NSs coupling with US irradiation, the changes of $H_2O_2$ concentration in the above reaction system were synchronously detected and recorded with degradation of MB. As shown in Fig. 4c, d, for these FeOCl NSs group, FeOCl/FeOOH NSs group, and FeOCl NSs + US group with extra added $H_2O_2$, the decreased $H_2O_2$ were identified with the generation of ·OH, which means the extra added $H_2O_2$ was the only source of ·OH. For FeOCl NSs + US group without extra added $H_2O_2$, there was also no $H_2O_2$ generation,

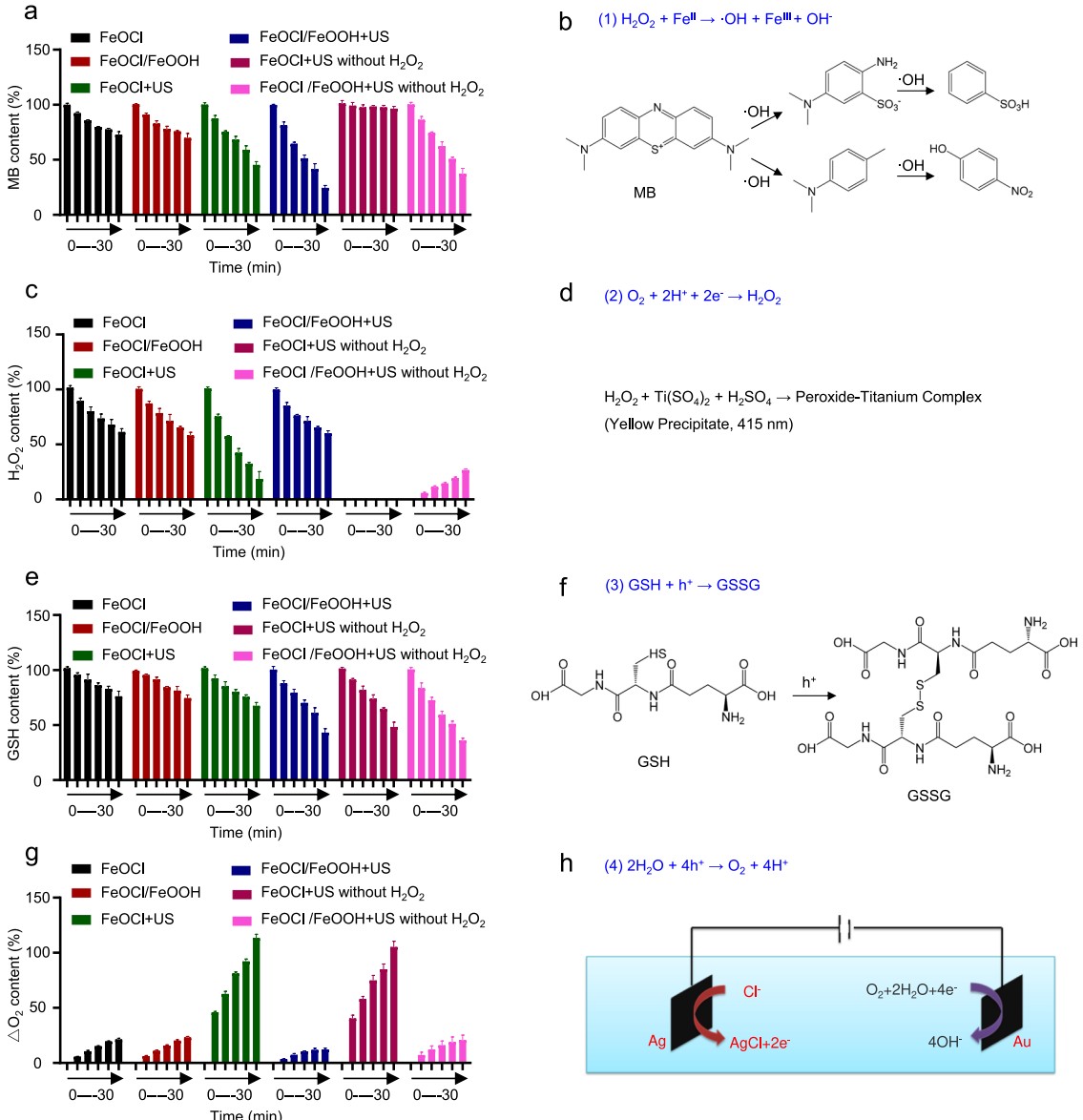

**Fig. 4 Catalytic performance analysis of FeOCl NSs and FeOCl/FeOOH NSs. a**, **b** The degradation of MB caused by FeOCl NSs and FeOCl/FeOOH NSs under different treatments. Data are presented as mean ± s.d. ($n = 3$ independent experiments). **c**, **d** The change of $H_2O_2$ caused by FeOCl NSs and FeOCl/FeOOH NSs under different treatments. Data are presented as mean ± s.d. ($n = 3$ independent experiments). **e**, **f** The degradation of GSH caused by FeOCl NSs and FeOCl/FeOOH NSs under different treatments. Data are presented as mean ± s.d. ($n = 3$ independent experiments). **g**, **h** The generation of $O_2$ caused by FeOCl NSs and FeOCl/FeOOH NSs under different treatments. Data are presented as mean ± s.d. ($n = 3$ independent experiments). The condition for US treatment is 1 MHz, 1 W cm$^{-2}$, 50% duty cycle. Source data are provided as a Source Data file.

demonstrating the FeOCl NSs exposed to the US cannot generate $H_2O_2$ due to unfavorable band position of FeOCl to generate $H_2O_2$. The significantly slow decreasing of $H_2O_2$ concentration in FeOCl/FeOOH NSs + US with extra added $H_2O_2$ group and the obvious generation of $H_2O_2$ in FeOCl/FeOOH NSs + US without extra added $H_2O_2$ group directly confirmed the excellent $H_2O_2$ self-supplying ability of FeOCl/FeOOH NSs coupling with US irradiation. Moreover, the $H_2O_2$ and ·OH production data catalyzed by FeOCl NSs and FeOCl/FeOOH NSs with or without US irradiation and with or without PEGylation were also detected and quantitative analysis. As shown in Supplementary Fig. 8a, b, the quantitative ·OH and $H_2O_2$ production data were identified with the data exhibited in Fig. 4a, c. In addition, under the same condition, FeOCl NSs and FeOCl/FeOOH NSs with PEGylation displayed higher catalytic activity compared to that without PEGylation.

The enhanced dispersibility endowed by PEGylation providing much more catalytic active sites should be the main reason. To understand the microscopic mechanism of the hydroxyl radical generation, we calculated the Gibbs free energy change (ΔG) of the $H_2O$ pathway and the $H_2O_2$ pathway, respectively, using DFT method. The ΔG is an effective parameter to character the catalytic activity, which follows the expression below,

$$\Delta G = \Delta E + \Delta E_{ZPE} - T\Delta S \qquad (1)$$

where ΔE, $\Delta E_{ZPE}$, T, and ΔS stand for the electronic energy difference, the zero-point energy difference, the human body temperature (310.15 K), and the entropy difference, respectively. The zero-point energy was obtained by the vibration frequency calculations. In a vibration frequency model, adsorbate species is released and the substrate is fixed due to the insignificant vibration of the substrate.

Two possible reaction pathways for the hydroxyl radical generation starting from $H_2O$ and $H_2O_2$, respectively, are listed below,

$$* + H_2O \rightarrow *OH + H^+ + e^- \qquad (a)$$

$$*OH \rightarrow * + \cdot OH \qquad (b)$$

$$* + H_2O_2 \rightarrow *OH + \cdot OH \qquad (c)$$

where * represents an adsorption site on the catalyst which is FeOCl phase in $H_2O$ pathway and FeOOH phase in $H_2O_2$ pathway, *OH represents the adsorbed OH on the catalytic site. In step (a), the free energy of the electron-proton pair was considered equal to that of $1/2$ $H_2$ at ambient conditions. In step (c), Fenton reaction product, Fe(III)-hydroxide pair was considered equal to *OH.

The schematic diagram of catalytic pathways for $\cdot OH$ generation is illustrated in Supplementary Fig. 9 and Supplementary Table 4. For the $H_2O$ pathway, one $H_2O$ molecule undergoes steps (a) and (b) successively to generate one $\cdot OH$ molecule. The $\Delta G$ of step (a) and (b) are 3.14 eV and $-0.27$ eV, respectively, meaning that the rate-limiting step to generate $\cdot OH$ from $H_2O$ is step (a) with an energy barrier of 3.14 eV. For $H_2O_2$ pathway, one $H_2O_2$ molecule generate one $\cdot OH$ molecule via step (c) with an energy barrier of 1.57 eV. Although the two proposed catalytic reaction pathways are endothermic processes, the $H_2O_2$ pathway has a much lower energy barrier and less energy requirement, so a more favorable $\cdot OH$ generation mechanism is that the $H_2O_2$ molecules are catalyzed by FeOOH via one-step Fenton reaction.

The redox homeostasis and hypoxia in TME bring about great obstacles for the effect of CDT and apoptosis of cancer cells[2,7,35]. Here, we intend to evaluate the capacities of FeOCl/FeOOH NSs to modulate the TME, including GSH consumption and $O_2$ generation. GSH, one of the most important keys to regulate intracellular REDOX equilibrium, was over-expressed in tumor cells. The intratumoral over-expressed GSH (0.32 eV oxidation potential of GSH/GSSG) was also oxidized by other oxidants, such as $Fe^{2+}/Fe^{3+}$ pairs and separated holes. As shown in Fig. 4e, f and Supplementary Fig. 10, for FeOCl NSs and FeOCl/FeOOH NSs groups, the ionization of $Fe^{2+}/Fe^{3+}$ could be the primary reason for the oxidation of GSH. For FeOCl NSs + US and FeOCl/FeOOH NSs + US groups, the excited holes with strong oxidated ability contributed to the other part of the oxidation of GSH. Due to the Fenton reaction recirculates $Fe^{3+}$, FeOCl NSs + US and FeOCl/FeOOH NSs + US groups with extra added $H_2O_2$ also accelerated the consumption of GSH. In addition, because of the high oxidizability of $Fe^{3+}$, a continuous $O_2$ production was observed in FeOCl NSs and FeOCl/FeOOH NSs with extra added $H_2O_2$ groups (Fig. 4g, h). Interestingly, the $O_2$ generation was greatly enhanced exposed to US irradiation. Coupling with the $O_2$ generation in NSs + US groups even without added $H_2O_2$, the US-excited holes catalyzed water splitting may be the main reason for the continued $O_2$ generation.

One of the most popular US-mediated catalysis is piezo-catalysis relied on piezoelectric materials[36–38]. A piezocatalytic reaction is generally due to piezoelectric polarization generated by piezoelectric materials triggering an electrochemical process. Piezoelectric materials with noncentrosymmetric structures possess electric dipoles, which will form an external electric field under mechanical deformation, such as US irradiation[39]. Hence, in order to dig the mechanism of FeOCl/FeOOH NSs mediated catalysis under US irradiation, the piezoelectric properties of FeOCl/FeOOH NSs were characterized in detail. To probe piezoelectric properties in FeOCl/FeOOH NSs, piezoresponse force microscopy (PFM) technique has been employed. A localized point-to-point piezo-response of FeOCl/FeOOH NSs has been probed by applying a probe bias of $\pm 8$ V. Supplementary Fig. 11a, b represent PFM amplitude and phase signals,

respectively. However, amplitude butterfly loop and phase hysteresis, the representative properties of piezoelectric materials, were not observed in Supplementary Fig. 11a, b, which illustrated there were no piezoelectric properties in FeOCl/FeOOH NSs and piezo-catalysis might not be the main mechanism behind FeOCl/FeOOH NSs. Another possible mechanism of this reaction is sonodynamic or sonocatalytic effect relied on sonosensitizers or sonocatalysts. The sonosensitizers or sonocatalysts are mainly semiconductors with suitable band structure, which are similar to photosensitizers or photocatalysts. When traveling through a liquid environment or tissue, the US will induce gas-filled microbubbles to oscillate in the acoustic field. This process is called cavitation and with the increase of the acoustic pressure, the microbubbles will finally implode. Upon implosion of the microbubbles, lots of heat and sometimes light (sonoluminescence) is released. Such an energy in a short period of time is able to trigger the separation of electron-hole pairs of sonosensitizers to generate ROS by catalyzing redox reaction with surrounding substrates. Hence, in order to deeply dip the mechanism of FeOCl/FeOOH NSs mediated $H_2O_2$ self-supplying under US irradiation, the band structure, including band gaps, conductive band levels, and valence band levels, and the electrostatic potentials, including work functions and Fermi levels, were detected and calculated detailly. Firstly, the band structure of FeOCl/FeOOH NSs was tested and calculated via diffuse reflection absorption and XPS spectra. As shown in Fig. 5a, b, due to the heterojunction structure, two curves of the diffuse reflection absorption of FeOCl/FeOOH NSs were exhibited. Calculating from their tangent lines, the band gaps (Eg) of FeOCl and FeOOH in FeOCl/FeOOH NSs were detected to be 1.8 and 2.1 eV, respectively. The XPS spectra supplied the VB values of FeOCl and FeOOH of FeOCl/FeOOH NSs, which calculated to be 2.7 and 2.2 eV, respectively (Fig. 5c). Then, the CB values of FeOCl and FeOOH were determined to be 0.9 and 0.1 eV, calculated from the difference between Eg and VB. As reported previously, this crossed band structure could induce type II or Z scheme charges transfer in the interface of this heterojunction[40–43]. For the type-II heterojunction, the photo/sono-excited electrons will aggregate transfer from the high CB of FeOOH to low CB of FeOCl, meanwhile, the photo/sono-excited holes will aggregate to low VB of FeOOH from high VB of FeOCl. As exhibited in Supplementary Fig. 12, the conventional type-II heterojunction-induced interfacial charge transfer would sharply damage the redox force of heterojunction[44]. In addition, the strong Coulomb electrostatic repulsive force between identical charges would hinder the type-II heterojunction-induced interfacial charge transfer. Z-scheme electron transfer is another potential electron transfer route. Basically, the excited electrons in the CB of FeOCl will be combined with the holes in the VB of FeOOH, maintaining stronger reduction/oxidation potentials of separated electrons and holes on the CB of FeOOH and the VB of FeOCl. Hence, this Z-scheme heterojunction seems to be more reasonable from viewpoint of kinetics and beneficial to the subsequent catalytic reaction, than type II heterojunction.

To further confirm the type of FeOCl/FeOOH NSs-based heterojunction, the interfacial charge transfer between FeOCl and FeOOH was further investigated by applying density functional theory (DFT) computational calculations. The electrostatic potentials of FeOCl and FeOOH were shown in Fig. 5d, e, in which the work functions of FeOCl and FeOOH were determined to be 6.68 and 4.83 eV, respectively. Due to the difference of work function, a potential charge transfer at the interface of FeOCl and FeOOH was constructed. The greater work function of FeOCl will give rise to the charge transfer from FeOOH to FeOCl until the Fermi level equilibrium. This charge transfer will lead to the formation of a built-in electric field at their interface. Hence,

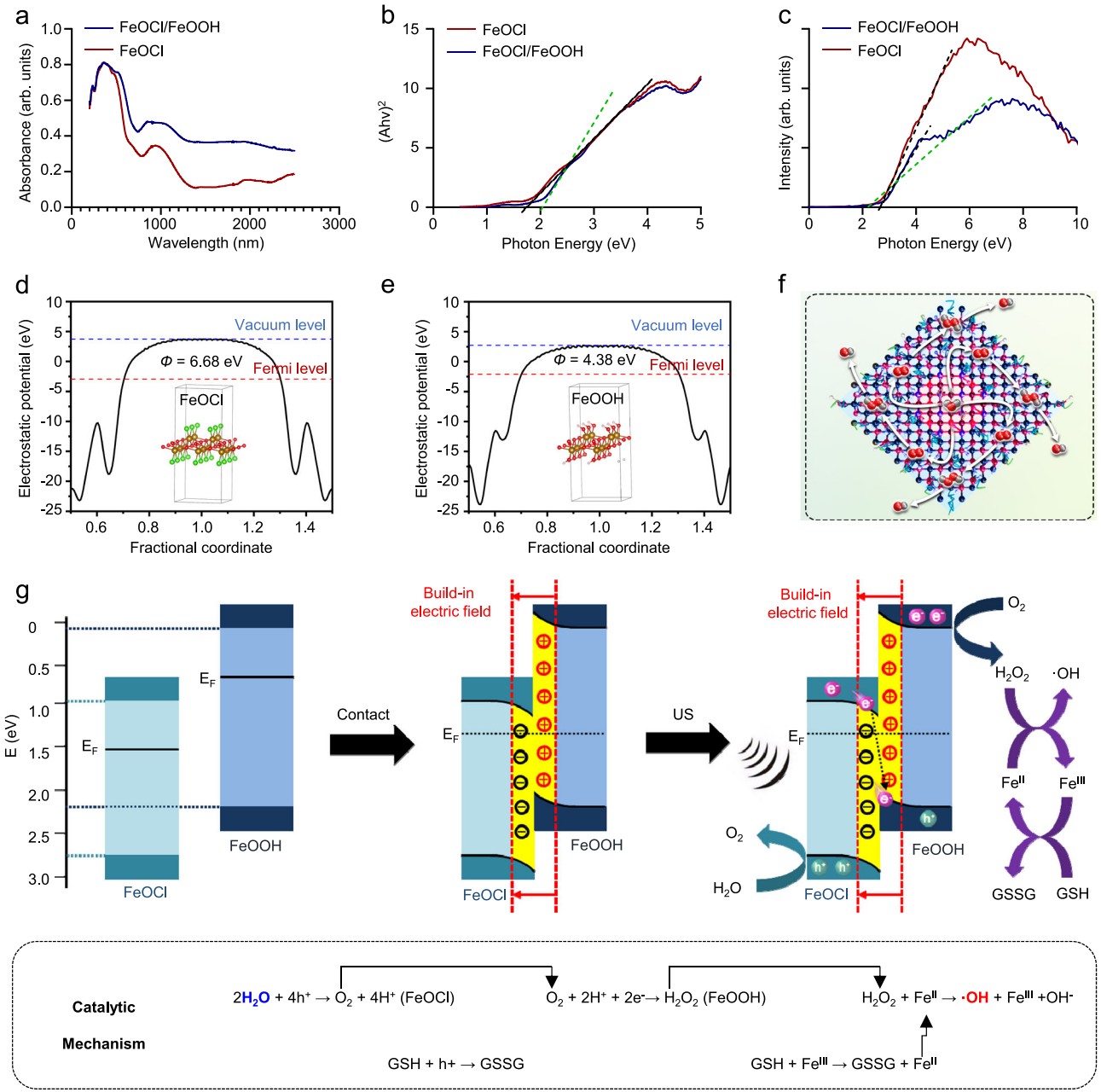

**Fig. 5 Catalytic mechanism analysis of FeOCl NSs and FeOCl/FeOOH NSs. a** UV–Vis absorbance spectra and **b** calculated band gap of FeOCl NSs and FeOCl/FeOOH NSs. **c** Valence band of FeOCl NSs and FeOCl/FeOOH NSs calculated from XPS spectra. Density functional theory (DFT) computational calculations of **d** FeOCl and **e** FeOOH. **f** Schematic illustration and chemical reaction equations of Z-schemed heterojunction based on FeOCl/FeOOH NSs for ·OH generation using water as the only substrate. **g** Mechanism of Z-schemed heterojunction based on FeOCl/FeOOH NSs for ·OH generation. For these characterizations of fabricated FeOCl NSs and FeOCl/FeOOH NSs, including UV–Vis absorbance spectra, band gap, valence band, and DFT computational calculations, three times each experiment was repeated independently with similar results. The units (arb. units) of UV–Vis absorbance spectra and XPS spectra mean arbitrary units. Source data are provided as a Source Data file.

the FeOOH is positively charged, while the FeOCl is negatively charged at the interface. This result is consistent with the charge transfer from FeOOH to FeOCl as revealed by XPS results. As observed in Fig. 3d–i, noticeably, the binding energy of Fe $2p$, Cl $2p$, and O $1s$ in FeOCl/FeOOH NSs were shifted by 0.2–0.5 eV toward lower binding energy in comparison with those of pristine FeOCl NSs, indicating that the electrons transferred from FeOOH to FeOCl upon hybridization due to the difference of their work functions and Fermi levels. Such electron transfer also demonstrated that a built-in electric field at their interfaces pointing from FeOOH to FeOCl was constructed, facilitating the

construction of Z-scheme FeOCl/FeOOH heterojunction without any redox mediator, which would efficiently separate the charge carriers and thus promote the reduction and oxidation reactions catalyzed separated electrons and holes. The built-in electric field facilitated Z scheme charges transfer in this FeOCl/FeOOH-based heterojunction was presented in Fig. 5g. In addition, the $E^0$ of oxidation of $H_2O/O_2$ (1.23 eV) and GSH/GSSG (0.3 eV) were lower than the VB of FeOCl. And the $E^0$ of reduction of $O_2/H_2O_2$ (0.69 eV) was lower than that of the CB of FeOOH. Hence, from a thermodynamic point of view, under US irradiation, it is feasible and convenient for $H_2O$ oxidation to produce $O_2$ via the holes in

the VB of FeOCl, and the produced $O_2$ is then reduced to $H_2O_2$ via the electrons in the CB of FeOOH, realizing the in situ $H_2O_2$ generation without any limitation from the tumor microenvironment (Fig. 5f, g).

**Antitumor strategy and biocompatibility in vitro**. Next, the TME-modulating and antitumor effect of FeOCl NSs and FeOCl/FeOOH NSs in vitro were investigated in detail. At first, the cytotoxicity of FeOCl NSs and FeOCl/FeOOH NSs to normal cells, such as human embryonic kidney cells (HEK293) and human normal liver cells (HL-7702) were detected. As exhibited in Fig. 6a, both the FeOCl NSs and FeOCl/FeOOH NSs exhibited good biocompatibility and safety to normal cells. By contrast, the FeOCl NSs and FeOCl/FeOOH NSs showed obviously specific cytotoxicity to cancer cells (Fig. 6b and Supplementary Fig. 13). The generated more cytotoxic OH via Fenton reaction resulted from high content of $H_2O_2$ in TME would be the primary cause for the specific cytotoxicity to cancer cells of FeOCl NSs and FeOCl/FeOOH NSs. A calcein acetoxymethyl ester (calcein-AM)/propidium iodide (PI) staining assay was performed to intuitively observe live and dead cells after various treatments. The results exhibited in Supplementary Fig. 14 showed obvious apoptosis in FeOCl NSs group, FeOCl/FeOOH NSs group, and FeOCl NSs + US group, while the maximum amount of apoptosis was observed in FeOCl/FeOOH NSs + US group. Next, for the TME-modulating capacity of FeOCl NSs and FeOCl/FeOOH NSs, the $O_2$-generating, $H_2O_2$-generating, and GSH-consuming performance in MCF7 cells was tested by using their corresponding assay kits. As shown in Fig. 6c, d, a rapid and FeOCl NSs and FeOCl/FeOOH NSs concentration-dependent GSH-consuming and $O_2$ evolution performance was exhibited, which attribute to the reaction between $Fe^{3+}$ in NSs and GSH or $H_2O_2$. In addition, the GSH-consuming ability was enhanced coupling with US irradiation, which might result from the excited holes oxidated GSH to GSSG. For the FeOCl NSs + US group, the US irradiation excited holes have a high oxidation potential for catalyzing $H_2O$ splitting with $O_2$ evolution from $H_2O$. Nevertheless, although the Z-scheme FeOCl/FeOOH NSs heterojunction promotes the separation of US-excited electrons and holes, which may accelerate $H_2O$ splitting and $O_2$ evolution, the $O_2$ concentration was much lower than that of the FeOCl NSs + US group. The main reason may be the reduction of $O_2$ to $H_2O_2$ by the US-excited electrons. Meanwhile, the $H_2O_2$ self-supplying ability of FeOCl NSs and FeOCl/FeOOH NSs in TME is shown in Fig. 6e. For the FeOCl NSs and FeOCl/FeOOH NSs groups, the decrease of $H_2O_2$ ascribed to the Fenton reaction mediated by NSs, which is also the major cause for the poor chemodynamic effect of Fenton reaction without replenishing of $H_2O_2$. For the FeOCl NSs + US group, the US irradiation accelerating Fenton reaction led to the fast consumption of $H_2O_2$. However, with increasing of NSs concentration and chemodynamic effect, the concentration of $H_2O_2$ nearly maintain a steady concentration, which further confirmed the excellent $H_2O_2$ self-supplying ability of FeOCl/FeOOH NSs under US irradiation.

As shown in Fig. 6b, FeOCl NSs and FeOCl/FeOOH NSs exhibited an obviously specific cytotoxicity for cancer cells, because of the relatively high $H_2O_2$ concentration in TME facilitated Fenton reaction to generate ·OH. As well-known and Fig. 6b shown, the ·OH generated by Fenton reaction with TME intrinsic $H_2O_2$ was insufficient to effectively induce apoptosis of tumor cells. Coupling FeOCl NSs with US irradiation, the ionization of $Fe^{2+}/Fe^{3+}$ was enhanced, inducing marginally high cytotoxicity than that of NSs groups. Interestingly, Z-scheme FeOCl/FeOOH NSs-based interplanar heterojunction exposed to US irradiation exhibited the highest cytotoxicity to cancer cells, in

which more than 90% tumor cells were dead with 100 μg/mL FeOCl/FeOOH NSs and 10 min US irradiation. The main cause of this excellent anti-tumor performance of FeOCl/FeOOH NSs should attribute to the supplying $H_2O_2$ through two steps cascade reaction ($H_2O \xrightarrow{h+} O_2 \xrightarrow{e-} H_2O_2$). To further test the influence of $H_2O_2$ concentration on the chemodynamic effect, a certain amount of $H_2O_2$ was slowly added to maintain the $H_2O_2$ concentration in TME in FeOCl NSs + US group. As shown in Fig. 6b, similar cytotoxicity of FeOCl NSs + US group was observed compared with FeOCl/FeOOH NSs + US group, which not only demonstrates the $H_2O_2$ concentration is vital for chemodynamic effect, but also confirm the excellent $H_2O_2$ supplying ability of FeOCl/FeOOH NSs. In addition, flow cytometry detection was also applied to test the cytotoxicity of FeOCl NSs and FeOCl/FeOOH NSs to cancer cells. Supplementary Fig. 15 and Fig. 6f exhibited a similar cytotoxicity and cancer cells apoptosis with MTT methods.

**ROS generation in vitro**. 2,7-Dichlorofluorescin diacetate (DCFH-DA) was applied as the intracellular ·OH detection probe for analyzing the performance of FeOCl/FeOOH NSs interplanar heterojunction-mediated CDT inducing ·OH burst. As shown in Fig. 7a, b, cancer cells treated with FeOCl NSs and FeOCl/FeOOH NSs alone showed a slightly ·OH production in contrast to control group, meaning FeOCl NSs and FeOCl/FeOOH NSs induced the weak Fenton reaction with intrinsic $H_2O_2$ in TME. FeOCl/FeOOH NSs plus US irradiation treatment triggered the strongest green fluorescence signal, due to the self-supplying $H_2O_2$ ability of interplanar heterojunction. The difference between FeOCl NSs + US group and FeOCl NSs + US + $H_2O_2$ group further demonstrated the significant role of $H_2O_2$ to CDT and the excellent self-supplying $H_2O_2$ ability of this interplanar heterojunction. The flow cytometry detection was also applied to test the intracellular ROS content after different treatments. Supplementary Fig. 16 and Fig. 7e exhibited a similar ROS content with that of confocal images. In addition, the hypoxia in TEM after different treatments was also detected. As shown in Fig. 7c, d, after treatment with FeOCl NSs and FeOCl/FeOOH NSs alone, a slight improvement of hypoxia was observed, attributing to the reverse Fenton reaction. For the FeOCl NSs + US group, a much weaker green fluorescence indicating the $O_2$-generating ability of the US-excited holes on the VB of FeOCl NSs. As a result of the $H_2O_2$ conversion from $O_2$ catalyzed by the US-excited electrons in the CB of FeOOH NSs, the content of $O_2$ was sharply decreased in FeOCl/FeOOH NSs + US group. It is well known that one of main route of ROS inducing apoptosis is triggering irreparable DNA damage[3,4,45]. Therefore, γ-H2AX, one of DNA damage probe, was employed to visually observe the irreparable DNA damage levels in MCF7 cells after different treatments. Figure 7f, g exhibited the treatment of FeOCl NSs and FeOCl/FeOOH NSs alone induced mild irreparable DNA damage in MCF7 cells. A little high levels of irreparable DNA damage in cancer cells was observed in the treatment of FeOCl NSs coupling with US irradiation. Moreover, coupling with the strong $H_2O_2$ self-supplying ability of FeOCl/FeOOH NSs-based interplanar heterojunction, a mass of irreparable DNA damage was exhibited in FeOCl/FeOOH NSs + US group. Moreover, a reliable marker for oxidative stress, 8-hydroxy-2′-deoxyguanosine (8-OHdG), was analyzed to further demonstrate this potential mechanism. Figure 7h displayed that there was a consistency between the 8-OHdG results and γ-H2AX data. These findings further demonstrated the highly efficient chemodynamic effect mediated by the Z-scheme interplanar heterojunction.

**Biodistribution and antitumor strategy in vivo**. Next, the in vivo therapeutic outcomes of FeOCl NSs and FeOCl/FeOOH NSs were further analyzed (Fig. 8a). At first, the in vivo

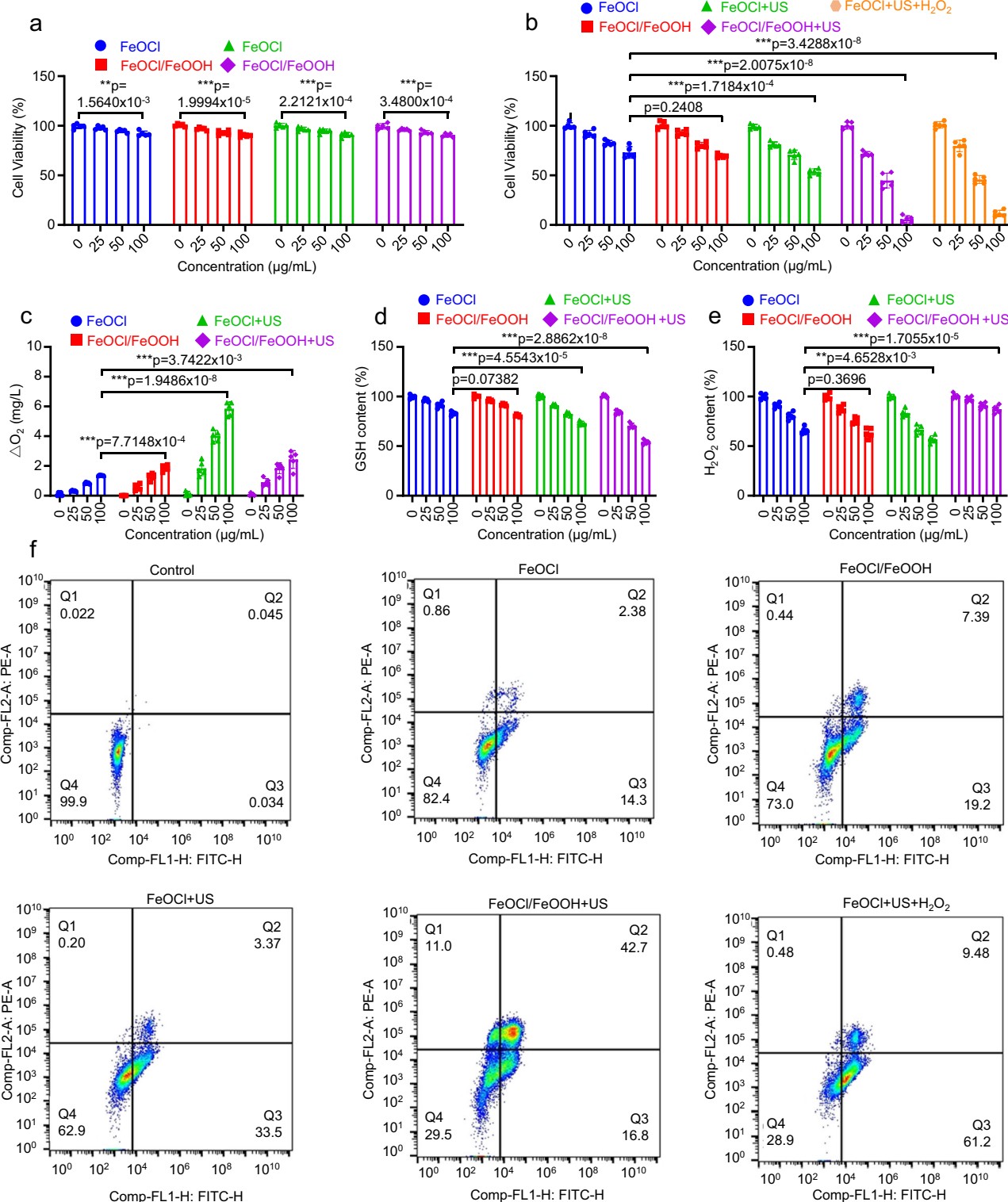

**Fig. 6 Biocompatibility and cytotoxicity of FeOCl NSs and FeOCl/FeOOH NSs. a** Relative viability of normal human cells (HL-7702 and HEK293) after incubation with FeOCl NSs or FeOCl/FeOOH NSs for 24 h. **b** Relative viability of MCF7 cells after incubation with FeOCl NSs or FeOCl/FeOOH NSs under different treatments for 24 h. **c** The generation of $O_2$ of MCF7 cells after incubation with FeOCl NSs or FeOCl/FeOOH NSs under different treatments. **d** The degradation of GSH of MCF7 cells after incubation with FeOCl NSs or FeOCl/FeOOH NSs under different treatments. **e** The consumption of $H_2O_2$ of MCF7 cells after incubation with FeOCl NSs or FeOCl/FeOOH NSs under different treatments. **f** FCM images of MCF7 cells after incubation with FeOCl NSs and FeOCl/FeOOH NSs under different treatments for 12 h. The condition for US treatment is 1 MHz, 1 W cm$^{-2}$, 50% duty cycle, 5 min. Data are presented as mean ± s.d. ($n = 5$ biologically independent cells). Statistical differences were analyzed by Student's two-sided $t$-test. Source data are provided as a Source Data file. The information on antibodies for these assays are provided in Supplementary Table 5.

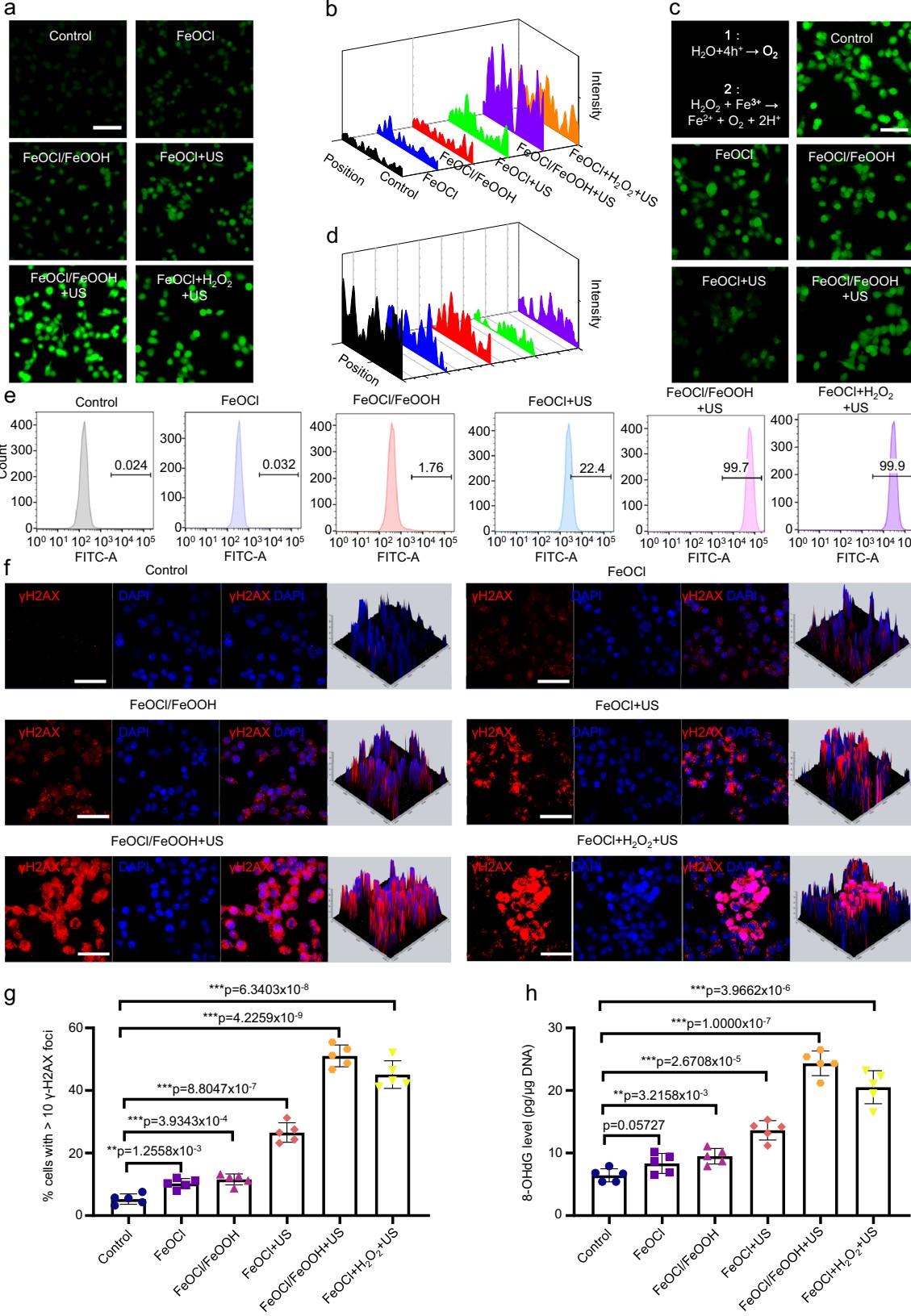

fluorescence imaging ability of FeOCl/FeOOH NSs was investigated using Cy7 loaded FeOCl/FeOOH NSs (Supplementary Fig. 17). As shown in Fig. 8b, the FeOCl/FeOOH NSs accumulated at the tumor site effectively and continuously. Then the mice were sacrificed and the tumors were dissected after 24 h post-injection. Figure 8b and Supplementary Fig. 18 showed the

different accumulation of NSs in normal organs and tumor. In addition, pharmacokinetic analysis declared the long circulation time of FeOCl/FeOOH NSs (Fig. 8c). Moreover, an inductively coupled plasma emission spectrometer (ICP) was applied to more precisely quantified the accumulation efficiency of FeOCl/FeOOH NSs in tumor tissue and main organs. As shown in Fig. 8d, after

**Fig. 7 Intracellular ROS generation, $O_2$ generation, and DNA damage of MCF7 cells under different treatments. a** The intercellular ROS generation and **c** the $O_2$ generation of MCF7 cells under different treatments with FeOCl NSs or FeOCl/FeOOH NSs, scale bar = 100 μm. For these laser scanning confocal microscopy images of ROS and $O_2$, three times each experiment was repeated independently with similar results. **b, d** Fluorescence quantitative analysis of intracellular ROS and $O_2$ generation. **e** Intracellular ROS generation detected by FCM. Three times each experiment was repeated independently with similar results. **f** Representative confocal microscopy images of the MCF7 cells (scale bars = 30 mm) after different treatments. Three times each experiment was repeated independently with similar results. The nuclei were stained by DAPI (blue), and the γH2AX foci per nucleus were stained by anti-γH2AX antibody (red). **g** Quantification of the percentages of cells with >10 γH2AX foci numbers by confocal microscopy. **h** In vitro DNA damage of the cells after different treatments were measured by 8-OHdG assay. The condition for US treatment is 1 MHz, 1 W cm$^{-2}$, 50% duty cycle, 10 min. Data are presented as mean ± s.d. ($n = 5$ biologically independent cells). Statistical differences were analyzed by Student's two-sided $t$-test. Source data are provided as a Source Data file.

24 h post-injection, a high tumor accumulation efficiency of 16.5% was detected. The high accumulation of drugs in the tumor site is the cornerstone of excellent antitumor performance. Meanwhile, the accumulation efficiency of FeOCl/FeOOH NSs in the liver reached 31%, which was caused by the absorption of the mononuclear phagocyte system (Fig. 8d).

Next, the in vivo therapeutic research of FeOCl NSs and FeOCl/FeOOH NSs were performed. After separating MCF7 tumor-bearing mice into six groups randomly, different treatments were carried out to each group: treatment 1: saline; treatment 2: US; treatment 3: FeOCl NSs; treatment 4: FeOCl/FeOOH NSs; treatment 5: FeOCl NSs + US; and treatment 6: FeOCl/FeOOH NSs + US. The dose of FeOCl NSs and FeOCl/FeOOH NSs intravenously injected into mice in treatments 3, 4, 5, and 6 were 5 mg/kg. The US treatment (1 MHz, 1 W cm$^{-2}$, 50% duty cycle) in treatments 5 and 6 were performed at 24 h post-injection of NSs (Supplementary Fig. 19). As shown in Fig. 8e–g, no significant tumor growth inhibition could be observed in the control (treatment 1) or US-only (treatment 2), but to a certain extent curative effect was exhibited in FeOCl NSs and FeOCl/FeOOH NSs alone treated mice (treatments 3 and 4). The production of ·OH through FeOCl NSs or FeOCl/FeOOH NSs mediating Fenton reaction and consumption of GSH might be the primary cause for this mild inhibition of tumor growth. Treatment 5 exhibited a better therapeutic effect compared to treatment 3, demonstrating the advantages of US irradiation for enhancing chemodynamic effect via improving ionization of $Fe^{2+}/Fe^{3+}$ in FeOCl NSs. Notably, treatment 6 obtained nearly complete elimination of tumors without recurrence due to the largely enhanced chemodynamic effect via FeOCl/FeOOH NSs-based interplanar heterojunction with $H_2O_2$ self-supplying ability. The tumor inhibition rate based on the average volume of tumors exhibited that the tumor inhibition rate of the treatment 6 group was 99.8% (Supplementary Fig. 20). The excellent therapeutic outcomes of interplanar heterojunction-mediated catalytic therapy were also exhibited in the tumor-bearing mice images (Supplementary Fig. 21). Correspondingly, FeOCl/FeOOH NSs coupled with US treatment were associated with a high survival rate (Fig. 8h). Moreover, all treatments showed nearly no side effects (Fig. 8i).

**ROS burst in vivo**. In order to further confirm the highly efficient catalytic performance, the ROS generation in vivo was investigated by DCFH-DA staining. Figure 9a, b exhibited similar ROS production detected in vitro. Specifically, a mild ROS accumulation after treatment with FeOCl NSs and FeOCl/FeOOH NSs alone. Treatment with FeOCl/FeOOH NSs + US showed a strong green fluorescence, indicating the highest chemodynamic efficacy. Next, Cleaved Caspase-3 (C-CAS3) and γ-H2AX, markers for cell apoptosis and DNA double-strand breaks, respectively, were employed to analyze the cell apoptosis and DNA damage levels in vivo. Due to limited substrates in TME, a certain extent of cancer cell apoptosis and irreparable DNA damage were observed in FeOCl NSs and FeOCl/FeOOH NSs-treated mice. Coupling with US irradiation, the tumor section exhibited remarkably high

levels of cell apoptosis and irreparable DNA damage, especially for FeOCl/FeOOH NSs (Fig. 9a, d). Furthermore, 8-OHdG was investigated to demonstrate this potential mechanism (Fig. 9c). Collectively, these above results demonstrated that the FeOCl/FeOOH NSs-based catalytic therapy could efficiently induce a ROS burst and specifically cancer cell apoptosis. Additionally, the largely enhanced chemodynamic effect of FeOCl/FeOOH NSs was also confirmed by TUNEL and HE staining.

**Biodegradability and biosafety**. Having demonstrated the effective tumor eradication induced by the 2D interplanar heterojunction FeOCl/FeOOH NSs, their biodegradability by dialysis in PBS, medium, and serum solutions were further investigated. The UV-vis-NIR absorbance of the residual FeOCl/FeOOH NSs left in the dialysis bag was detected every 2 days during the 14 days dialysis to evaluate the degradation rate. As shown in Fig. 10a–c, very slow biodegradation behaviors of FeOCl/FeOOH NSs were observed in PBS and medium solutions, demonstrating a good storage stability. A relatively fast biodegradability behavior of FeOCl/FeOOH NSs was exhibited in serum, which illustrated a desired biodegradability in vivo. To further reveal the biodegradation mechanism of FeOCl/FeOOH NSs, TEM images and size distributions of FeOCl/FeOOH NSs after 14 days in different solutions were detected, and shown in Fig. 10d, e. A kind of degradation phenomenon spreading from edge to center was observed in their TEM images. The gradually decreased size distribution also demonstrated the biodegradability of FeOCl/FeOOH NSs. Moreover, the biodegradability of FeOCl/FeOOH NSs in vivo were further analyzed by ICP. As shown in Fig. 10f, the accumulated FeOCl/FeOOH NSs in normal organs and tissues could be gradually degraded and excreted by the body over time. As shown in Supplementary Fig. 22, no sign of normal organ injury was detected, demonstrating the excellent in vivo biosafety of the prepared FeOCl/FeOOH NSs. The biocompatibility of FeOCl/FeOOH NSs was further confirmed through hematological, and immunological results. For immune analysis, the serum level of IL-6, TNF-α, and IFN-γ at 2 h and 24 h post intravenous injection of FeOCl/FeOOH NSs was measured. As exhibited in Supplementary Fig. 23, all cytokine levels from the FeOCl/FeOOH NSs-treated group showed no statistically significant differences compared with the control group, confirming that the prepared FeOCl/FeOOH NSs have good biocompatibility and biosafety in vivo. Besides, there were no obvious differences in biochemical detection detected between the mice of control group and those administrated with FeOCl/FeOOH NSs for 1, 7, and 14 days in the blood parameters of albumin (ALB), aminotransferase (ALT), alanine aspartate aminotransferase (AST), blood urea nitrogen (BUN), creatinine (Cr), total protein (TP), C-reactive protein (CRP), lactate dehydrogenase (LDH), amylase (AMY), creatine kinase (CK), and γ-glutamyl transpeptidase (γ-GT) (Fig. 10g). Moreover, the hematological detection was further carried out for evaluating the biocompatibility of FeOCl/FeOOH NSs. Red blood cells (RBC), white blood cells (WBC), hematocrit

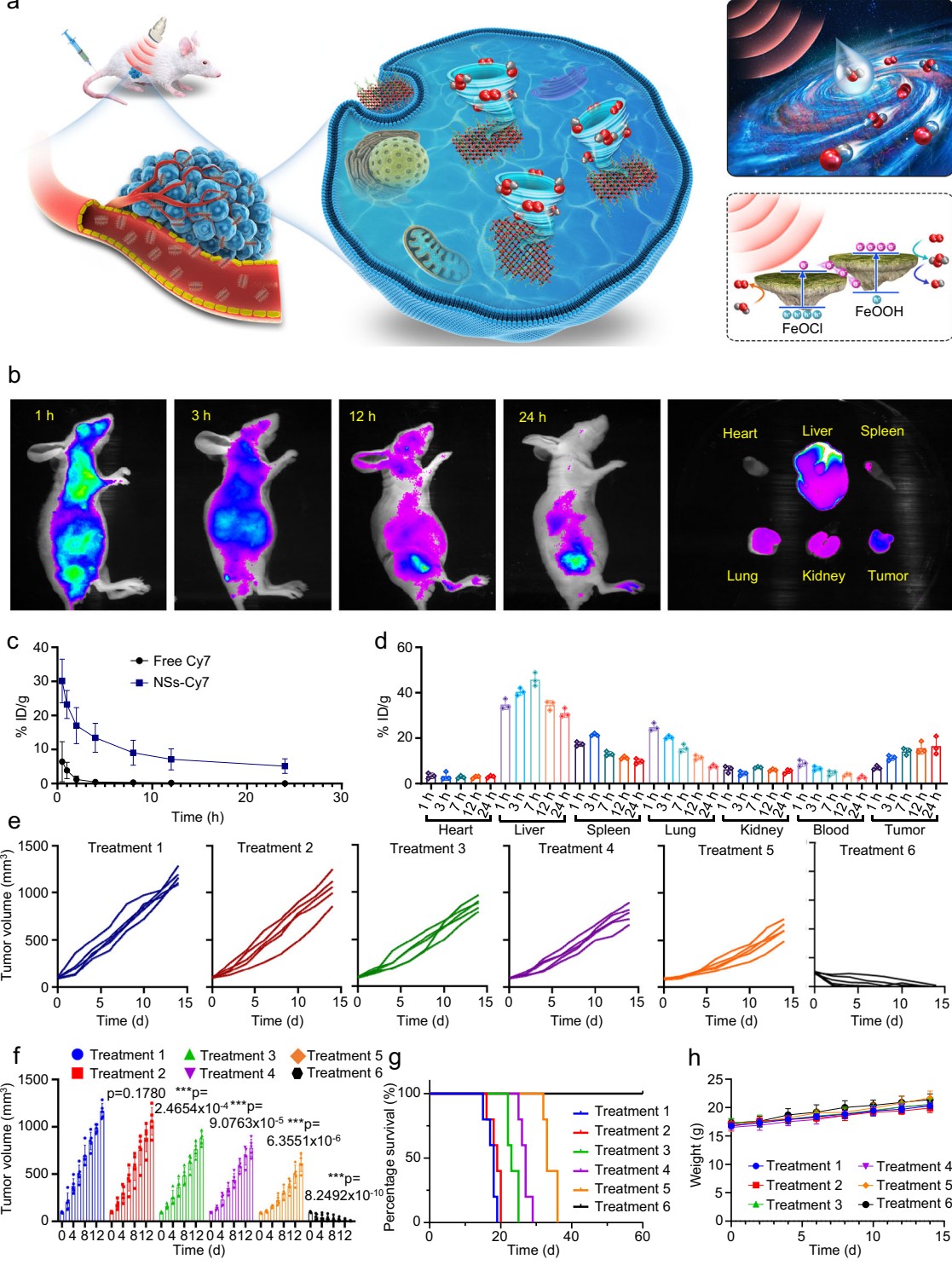

**Fig. 8 In vivo imaging and anti-tumor performance of FeOCl NSs and FeOCl/FeOOH NSs. a** Schematic diagram of treatment. **b** In vivo fluorescence images of nude mice after i.v. administration of NSs, and the ex vivo fluorescence images of the tumor and major organs at 24 h post-injection of Cy7 loaded FeOCl/FeOOH NSs. Data are presented as mean ± s.d. ($n = 3$ biologically independent mice). **c** Blood circulation of FeOCl/FeOOH NSs. Data are presented as mean ± s.d. ($n = 3$ biologically independent mice). **d** Bio-distribution of FeOCl/FeOOH NSs under different times. Data are presented as mean ± s.d. ($n = 3$ biologically independent mice). **e** Tumor growth curves of MCF7 tumor-bearing nude mice. **f** Tumor growth curves of MCF7 tumor-bearing nude mice. Data are presented as mean ± s.d. ($n = 5$ biologically independent mice). Statistical differences were analyzed by Student's two-sided $t$-test. **g** Survival rate of mice undergoing different treatments. **h** Body weight of mice during treatment. Data are presented as mean ± s.d. ($n = 5$ biologically independent mice). The condition for US treatment is 1 MHz, 1 W cm$^{-2}$, 50% duty cycle, 10 min. Source data are provided as a Source Data file.

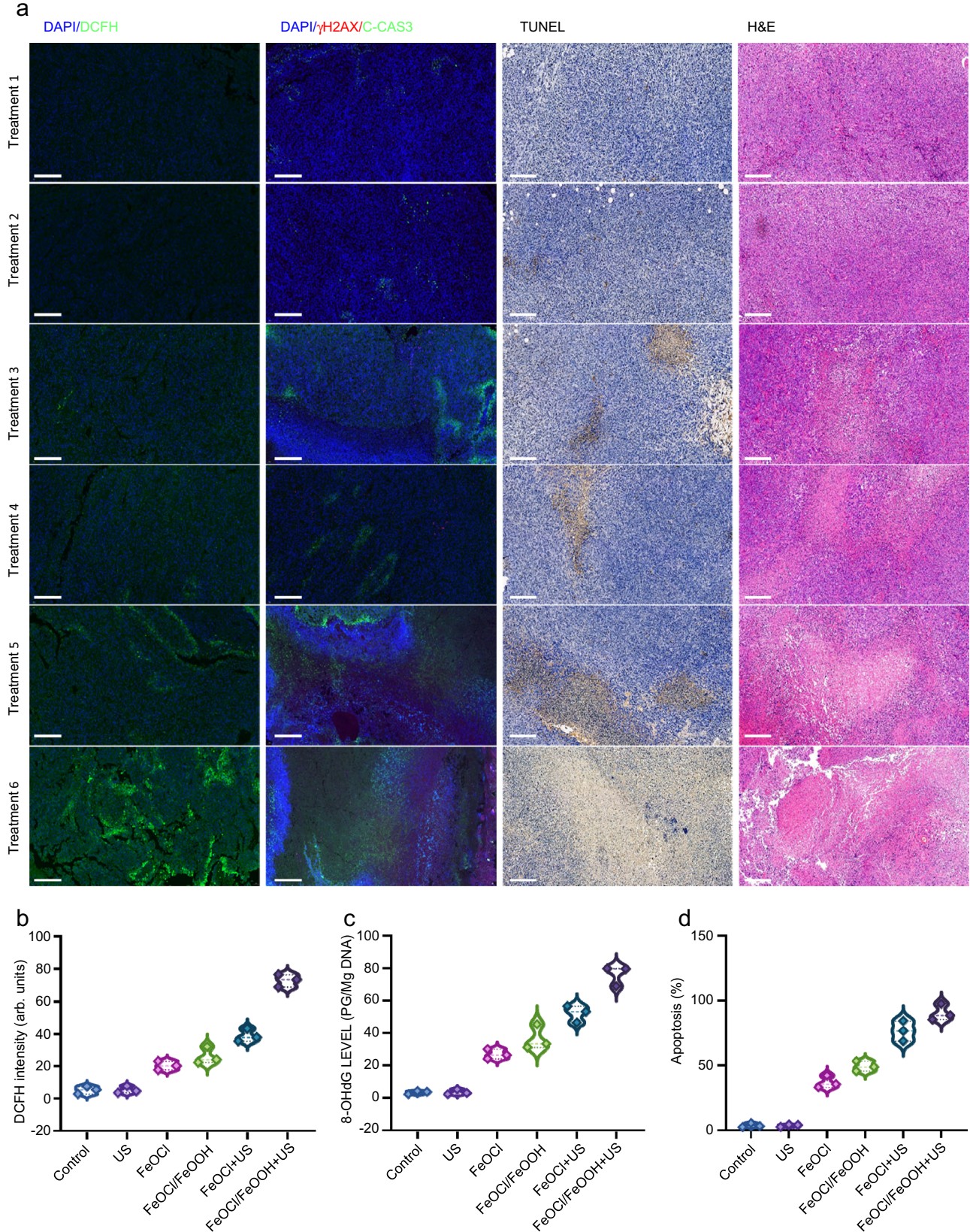

(HCT), hemoglobin (HGB), mean corpuscular hemoglobin (MCH), mean corpuscular hemoglobin concentration (MCHC), mean corpuscular volume (MCV), and platelet (PLT) counts were

measured (Fig. 10h). Compared with the control group, although there were a little increase in some indicators, no statistically significant difference of the FeOCl/FeOOH NSs-treated groups

**Fig. 9 In vivo immunofluorescence staining and analysis of different treatments. a** In vivo ROS generation, DNA damage, apoptosis detection in the sections from tumors. The generated ROS was stained by dichlorodihydrofluorescein (DCFH) (green). Three times each experiment was repeated independently with similar results. The nucleus is stained by DAPI (blue), damaged DNA by γH2AX foci (red), and apoptotic cells by apoptosis marker C-CAS3 (green). **b** Quantification of the in vivo ROS signals from the tumors under different treatments. Data are presented as mean ± s.d. ($n = 3$ biologically independent mice). **c** In vivo DNA damage of the tumors after different treatments was measured by 8-OHdG assay. Data are presented as mean ± s.d. ($n = 3$ biologically independent mice). **d** In vivo apoptosis of the tumor cells after different treatments. Data are presented as mean ± s.d. ($n = 3$ biologically independent mice). Source data are provided as a Source Data file. The information on antibodies for these assays are provided in Supplementary Table 5.

with PBS-treated groups in all the parameters after i.v. injection for 7 and 14 days. Supported by the above results, FeOCl/FeOOH NSs were regarded as relatively biocompatible and biosafe in vivo.

## Discussion

Cancer is one of the biggest killers of human health[46]. Catalysis, a versatile "tool", has created a colorful world for human beings' survival and development over the past several centuries. In the past decade, cross-amalgamation catalysis and medicine, named catalytic medicine or catalytic therapy, have integrated catalytic technology to solve medical problems, providing effective therapeutic strategies for a variety of diseases, especially cancer[47]. According to different catalytic mechanisms, catalytic therapy is divided into two categories: exogenous excitation catalysis and nonexogenous excitation catalysis[48]. Among them, exogenous excitation catalytic therapies mainly refer to the catalytic reactions triggered by specific external stimuli, such as light and US. Photodynamic therapy is the most representative light-triggered catalytic therapy, which severely limited by the poor penetration of light. US has been broadly used in clinic for disease diagnosis and treatment based on its noninvasiveness, minor energy attenuation, and high tissue-penetrating capability, such as US imaging, sonodynamic therapy, and piezocatalytic therapy[49]. Piezocatalytic therapy relying on the piezoelectric polarization generated by piezoelectric materials triggering an electrochemical process, holds great potential in tumor therapy and other biomedical fields. However, the high demand for piezoelectric materials has limited the development of piezocatalytic therapy. Sonodynamic or sonocatalytic therapy mainly refers to the electron-hole pair separation of catalysts under US irradiation, and use the redox capacity of the separated electrons or holes to catalyze a certain redox reaction to produce therapeutic products. The broad spectrum of the catalyst and the deep penetration of US give sonodynamic/sonocatalytic therapy unparalleled advantages.

As one of the most representative nonexogenous excitation catalysis, CDT, applying relatively high expression of $H_2O_2$ in tumor cells as target and substrate for production of highly cytotoxic ·OH, is a burgeoning and promising cancer treatment strategy[1–4]. Nevertheless, due to the poor catalytic activity of traditional CDT agents and limited substrate concentration, currently, CDT alone cannot achieve satisfactory efficacy, and is often used as a complementary therapy combined with other therapies. During past decades, despite enormous struggles, engineering CDT agents with high catalytic activity, specific and efficient $H_2O_2$-supplying functionality without being limited to the TME remains a great challenge.

In this work, we presented a collaborative combination, in which 2D interplanar heterojunction FeOCl/FeOOH NSs with high Fenton reaction catalyzing activity and US-triggering $H_2O_2$ supplying ability were fabricated and created a clever combination between CDT and sonodynamic/sonocatalytic therapy. Firstly, a smart wet-chemical method based on alkali replacement reaction was developed to intelligently integrate interplanar heterojunction synthesis and two-dimensional ultrathin

heterojunction exfoliation in one step. After PEGylation, the engineering FeOCl/FeOOH NSs functionalized with a characteristic interplanar heterojunction for efficient $H_2O_2$ self-supplying applying $H_2O$ as the only substrate was developed and exhibited a highly efficient chemodynamic effect. In the interplanar heterojunction FeOCl/FeOOH NSs, FeOCl part and FeOOH part with different Fermi levels and energy band structures contacting with each other induces charges to redistribute at their interfaces on account of the aligning Fermi levels, which mediates the construction of a built-in electric field at their interface. Under US irradiation, the US-excited electrons on the CB of FeOCl will be combined with the holes on the VB of FeOOH guided by the built-in electric field at their interface, leaving stronger reduction/oxidation potentials of separated electrons and holes on the CB of FeOOH and the VB of FeOCl. Meanwhile, a Schottky barrier was formed owing to band bending at their interface, which will prevent the electron flow from FeOOH to FeOCl and enhanced their Z-schemed charges transfer. Hence, a built-in electric field and Schottky barrier facilitated Z-schemed catalytic mechanism was constructed, in which the holes on the VB of FeOCl have a great ability to oxidize $H_2O$ and produce $O_2$, meanwhile, the generated $O_2$ will be immediately and directly reduced to $H_2O_2$ by the electrons on the CB of FeOOH. The self-supplying $H_2O_2$ guarantees the continuously and endlessly ·OH generation via the disproportionated reaction catalyzed by both FeOCl and FeOOH. The obtained FeOCl/FeOOH NSs with highly efficient chemodynamic effects exhibited excellent anti-cancer performance both in vitro and in vivo. Therefore, this research not only defines an intelligent strategy for the intelligent synthesis of 2D ultrathin heterojunction, but also gives evidence of the proof-of-concept application of the engineering 2D NSs in biomedical applications, which may also provide a certain incentive to make valuable contributions in other possible fields in the future. Since this 2D interplanar heterojunction uses $H_2O$ as the only needed exogenous substrate and US exposure triggering $H_2O_2$ and ·OH generation guarantees the targeting and specificity and this catalytic therapy, 2D interplanar heterojunction FeOCl/FeOOH NSs mediated catalytic therapy should possess huge potentials in other biomedical fields, such as sterilization and anti-infection of the diabetic wound, operative wound, and other infections.

## Methods

**Materials.** Ferric chloride (FeCl₃), sodiun hydroxide (NaOH), 2,7-dichlorodihydrofluorescein diacetate (DCFH-DA), methylene blue (MB), [Ru(dpp)₃]Cl₂ (RDPP), glutathione, 5,5′-dithiobis (2-nitrobenzoic acid) (DTNB), and H₂O₂ (30%) were supplied by Sigma-Aldrich. PEG-NH₂ (MW: 5k) and Cy7-PEG-NH₂ (MW: 5k) were supplied by Nanocs Inc. Trypsin-EDTA, phosphate buffer saline (PBS, pH 7.4), fetal bovine serum (FBS), RPMI medium, and DMEM medium were purchased from Gibco Life Technologies. 8-OhdG DNA Damage Quantification Direct Kit was purchased from EpiQuik™ Colorimetric. C-CAS3 (Asp175) and Anti-H2AX (pS139) antibodies were purchased from BD Pharmingen™.

**Preparation of FeOCl NSs and FeOCl/FeOOH NSs.** Bulk FeOCl powder was prepared by hydrothermal process using autoclave containing FeCl₃·6H₂O. After reaction at 180 °C for 24 h, probe sonication-assisted liquid exfoliation was employed for 12 h. After the exfoliation, the unexfoliated FeOCl was removed by

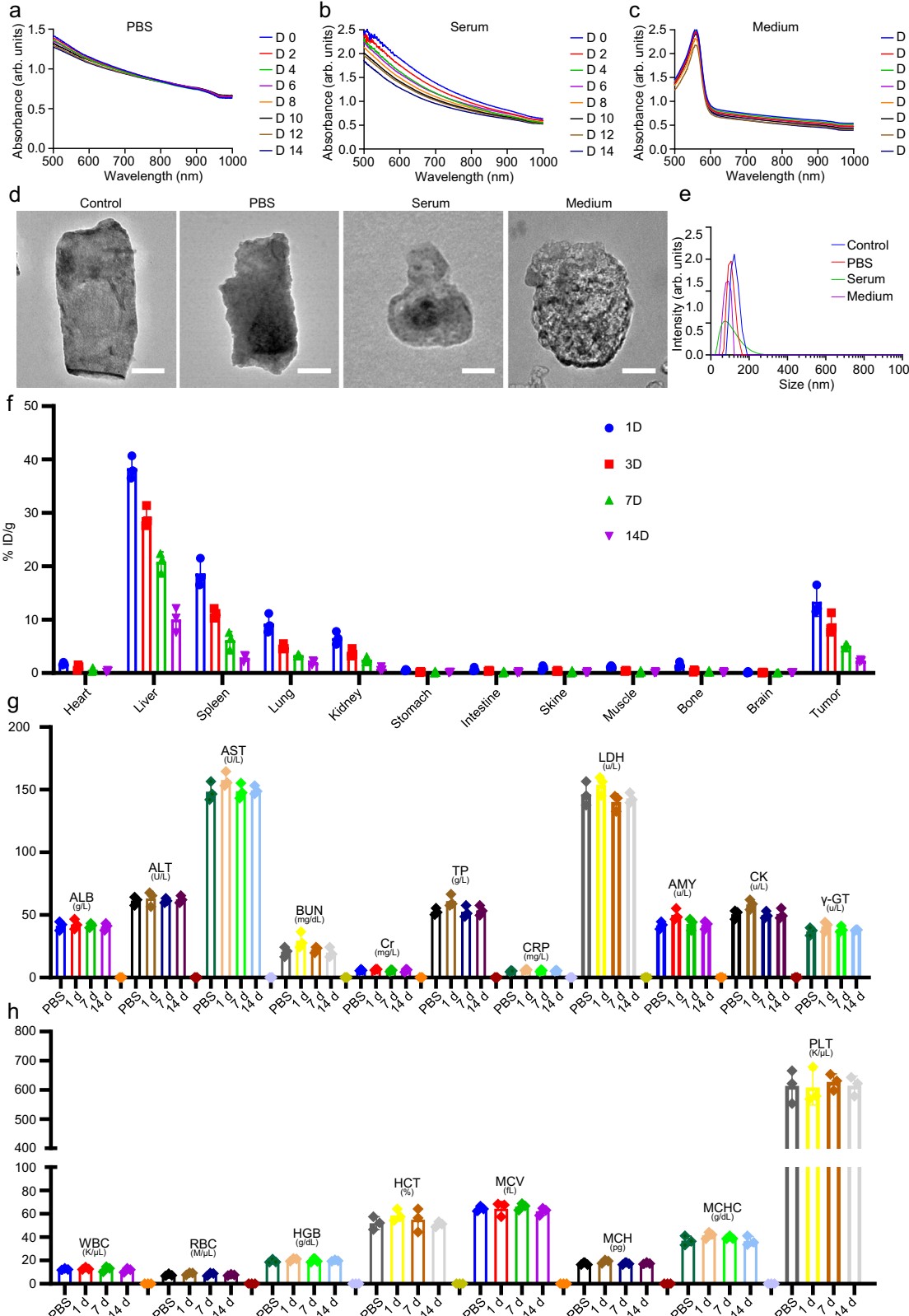

**Fig. 10 Biodegradation behavior of FeOCl/FeOOH NSs.** UV-vis-NIR absorbance spectra of layered FeOCl/FeOOH NSs co-incubated with **a** PBS, **b** serum and **c** medium solution for 14 days. The units (arb. units) of UV–Vis absorbance spectra and XPS spectra mean arbitrary units. **d** TEM images of layered FeOCl/FeOOH NSs and FeOCl/FeOOH NSs co-incubated with PBS, serum, and medium solution for 14 days, scale bar = 50 nm. For these TEM images of FeOCl/FeOOH NSs, three times each experiment was repeated independently with similar results. **e** Size distribution of layered FeOCl/FeOOH NSs and FeOCl/FeOOH NSs co-incubated with PBS, serum, and medium solution for 14 days. **f** Biodistribution of FeOCl/FeOOH NSs in tumor-bearing mice by ICP measurement. The data show mean ± s.d., $n = 3$ biologically independent mice. **g** Blood biochemistry analysis of Balb/c mice treated with PBS or FeOCl/FeOOH NSs. **h** Blood hematology analysis of Balb/c mice treated with PBS or FeOCl/FeOOH NSs. Data are presented as mean ± s.d. ($n = 3$ biologically independent mice). Source data are provided as a Source Data file.

centrifugation at $250 \times g$ for 5 min. Afterwards, the solution was centrifuged at $1006 \times g$ for 5 min and the supernatant was collected and stored at 4 °C.

For FeOCl/FeOOH NSs preparation, the prepared bulk FeOCl powder was firstly immersed in NaOH solution (0.5 mM) and bath sonicated for 2 h. Then the NaOH etched FeOCl powder were under probe sonication-assisted liquid exfoliation in water for 5 h. After the exfoliation, the unexfoliated FeOCl/FeOOH was removed by centrifugation at $250 \times g$ for 5 min. Afterwards, the solution was centrifuged at $1006 \times g$ for 5 min and the supernatant was collected and stored at 4 °C.

**PEGylation of FeOCl NSs and FeOCl/FeOOH NSs**. In order to improve the stability and biocompatibility of FeOCl NSs and FeOCl/FeOOH NSs, NSs were further modified with PEG(5k)-NH$_2$. First, PEG(5k)-NH$_2$ (10 mg) was added into NSs suspension and then the suspension was ultrasonicated for 30 min. After stirred for 12 h, the mixture was washed three times by centrifugation at $210 \times g$ (4 °C) for 30 min. Afterwards, the PEGylated NSs were resuspended in PBS stored at 4 °C for further use. To prepare Cy7-labeled FeOCl NSs and FeOCl/FeOOH NSs, The Cy7-labed PEG(5k)-NH$_2$ (Cy7-PEG(5k)-NH$_2$) and PEG(5k)-NH$_2$ (1:4) were then coated on the surface of NSs to generate Cy7-labeled NSs using the aforementioned method.

**Characterization**. Scanning electronic microscopy (SEM, JSM-6700F, JEOL, Japan), transmission electron microscopy (TEM, JEM-2100UHR, JEOL, Japan), and atomic-force microscopy (AFM, FASTSCANBIO, Germany) were applied to test the microstructure and morphology of prepared FeOCl and FeOCl/FeOOH NSs. Fourier transform infrared spectrophotometry (FT-IR, Nexus 470, Nicolet, Madison, WI, USA) and X-ray photoelectron spectroscopy (XPS, ESCA-LAB 250Xi, Japan) were used to analyze the chemical composition. X-ray diffraction (XRD) spectra, Raman shift spectra, and energy-dispersive X-ray spectroscope (EDS) (Inca X-MAX, Oxford, UK) were used to analyze the elementary composition and chemical structures of NSs. UV-Vis-NIR spectra of NSs were recorded by an Infinite M200 PRO spectrophotometer at room temperature.

**Modeling and calculation details**. Spin-polarized density functional theory (DFT) calculations on FeOCl and FeOOH were performed using VASP code[50]. The projector augmented wave (PAW) approach was adopted to deal with the electron-ion interactions[50,51]. Electron exchange-correlation terms were described by the Perdew-Burke-Ernzerhof (PBE) formalism of the generalized gradient approximation (GGA)[52]. The cut-off energy of 500 eV sufficed to represent the expansion of plane-wave. The DFT-D2 method was used to describe the van der Waals interactions[53]. To appropriately describe the on-site Coulomb repulsion for the structure with strongly correlated $d$-electrons, the Coulomb interaction ($U$) and exchange interaction ($J$) parameters for Fe atoms were set to be 4 and 1 eV, respectively[54]. The convergence threshold was set to be $1 \times 10^{-5}$ Ha in energy and $1 \times 10^{-2}$ Ha/Å in force. The $2 \times 2 \times 1$ super-cells of FeOCl and FeOOH were repeated periodically in the $x$-$y$ plane, while a vacuum layer of 15 Å was added along the $z$ direction to eliminate inter-layer interference. The Brillouin zone integration was sampled by $8 \times 6 \times 1$ Monkhorst-Pack $k$-points meshes[55]. To compare the affinity of FeOCl and FeOOH toward H$_2$O, the adsorption of H$_2$O on FeOCl and FeOOH were investigated. The initial binding sites for H$_2$O were searched by Monte Carlo (MC) annealing simulations, which allow a rotatable molecule to randomly translate on the surface of the substrate until the local energy minima reached[56,57].

The work function ($\Phi$) indicates the minimum energy required for an electron from the surface of a material and it follows the equation[58].

$$\Phi = E_{vac} - E_F$$

where $E_{vac}$ and $E_F$ represent the electrostatic potential of vacuum level and Fermi level, respectively.

To determine the binding ability between the substrates and H$_2$O, the adsorption energy ($E_{ad}$) was calculated with the following expression[59].

$$E_{ad} = E_{com} - E_{sub} - E_{H_2O}$$

where $E_{com}$, $E_{sub}$ and $E_{H_2O}$ are the total energy of adsorption complex, isolated substrate (FeOCl or FeOOH) and isolated H$_2$O molecule, respectively. A negative magnitude of $E_{ad}$ indicates exothermic adsorption process.

Bader's charge analysis was performed to estimate the charge transfer ($\Delta Q$) between nanosheet and H$_2$O. $\Delta Q$ can be defined as[60].

$$\Delta Q = Q_1 - Q_0$$

where $Q_0$ and $Q_1$ are the number of electrons occupied by the H$_2$O before and after adsorption, respectively. Based on the definition, a negative magnitude of $\Delta Q$ infers the electrons flow from the H$_2$O molecule to the substrate.

**GSH degradation in vitro**. First, GSH solution (0.1 mM) was mixed with DTNB (0.2 mg/mL). Then, the above solution was treated as following treatments: (1) FeOCl NSs, (2) FeOCl/FeOOH NSs, (3) FeOCl NSs + US, (4) FeOCl/FeOOH NSs + US, (5) FeOCl NSs + US with H$_2$O$_2$, and (6) FeOCl/FeOOH NSs + US with

H$_2$O$_2$. The final concentration of NSs was 0.1 mg/mL. The condition for US treatment is 1 MHz, 1 W cm$^{-2}$, 50% duty cycle. The final concentration of H$_2$O$_2$ in treatment 5 and 6 were 0.05 mM. During 30 min reaction, the absorbance of DTNB was recorded every 5 min using UV–vis spectroscopy.

**Extracellular O$_2$ production**. The O$_2$ production performance was detected as the following group: (1) FeOCl NSs, (2) FeOCl/FeOOH NSs, (3) FeOCl NSs + US, (4) FeOCl/FeOOH NSs + US, (5) FeOCl NSs + US without H$_2$O$_2$, and (6) FeOCl/FeOOH NSs + US without H$_2$O$_2$. The final concentration of NSs was 0.1 mg/mL. The condition for US treatment is 1 MHz, 1 W cm$^{-2}$, 50% duty cycle. The final concentration of H$_2$O$_2$ in treatment 1, 2, 3, and 4 were 0.05 mM. During 30 min reaction, the production of O$_2$ was recorded every 5 min using a dissolved oxygen meter.

**H$_2$O$_2$ generation in vitro**. Hydrogen peroxide assay kit was applied to detected the change of H$_2$O$_2$ in vitro. The H$_2$O$_2$ production performance was detected as the following group: (1) FeOCl NSs, (2) FeOCl/FeOOH NSs, (3) FeOCl NSs + US, (4) FeOCl/FeOOH NSs + US, (5) FeOCl NSs + US without H$_2$O$_2$, and (6) FeOCl/FeOOH NSs + US without H$_2$O$_2$. The final concentration of NSs was 0.1 mg/mL. The condition for US treatment is 1 MHz, 1 W cm$^{-2}$, 50% duty cycle. The final concentration of H$_2$O$_2$ in treatment 1, 2, 3, and 4 were 0.05 mM. During 30 min reaction, the concentration of H$_2$O$_2$ was recorded every 5 min using microplate reader following the hydrogen peroxide assay kit protocol.

**·OH generation in vitro**. Methylene blue (MB) was applied as ·OH indicator to detected the generation of FeOCl NSs and FeOCl/FeOOH NSs. The ·OH production performance was detected as the following group: (1) FeOCl NSs, (2) FeOCl/FeOOH NSs, (3) FeOCl NSs + US, (4) FeOCl/FeOOH NSs + US, (5) FeOCl NSs + US without H$_2$O$_2$, and (6) FeOCl/FeOOH NSs + US without H$_2$O$_2$. The final concentration of NSs was 0.1 mg/mL. The condition for US treatment is 1 MHz, 1 W cm$^{-2}$, 50% duty cycle. The final concentration of H$_2$O$_2$ in treatment 1, 2, 3, and 4 were 0.05 mM. During 30 min reaction, the absorbance of MB was recorded every 5 min using UV–vis spectroscopy.

**Biocompatibility of NSs in vitro**. Human normal liver cells (HL-7702, catalog number:77402), human embryonic kidney cells (HEK 293, catalog number: CRL-1573), human breast cancer cell (MCF7, catalog number: HTB-22) and human prostatic cancer cells (PC3, catalog number: CRL-1435) were obtained from the American Type Culture Collections (ATCC). The biocompatibility of FeOCl NSs and FeOCl/FeOOH NSs was tested using human normal cell lines, including HL-7702 and HEK293. Briefly, HL-7702 and HEK293 cells were seeded into two 96-well plates at a density of 5000 cells in every well and incubated for 24 h respectively. Afterwards, FeOCl NSs and FeOCl/FeOOH NSs (0–100 μg/mL) were added to the normal cells mentioned above and co-incubated for another 24 h. Finally, cell viabilities were detected by MTT assay.

**Consumption of GSH in cells**. The intracellular consumption of GSH was detected by GSH assay kit. Briefly, MCF7 cells were seeded into 96-well plates for 24 h (37 °C, 5% CO$_2$). Then, the cells were treated as following: (1) FeOCl NSs, (2) FeOCl/FeOOH NSs, (3) FeOCl NSs + US, and (4) FeOCl/FeOOH NSs + US. The final concentration of NSs was 0.1 mg/mL. The condition for US treatment is 1 MHz, 1 W cm$^{-2}$, 50% duty cycle, 5 min. After another 24 h incubation, the treated MCF7 cells were washed three times by PBS and collected for detection. Then, the obtained cells suspended in 1 mL PBS were crushed by an ultrasound cell crusher. Finally, the intracellular GSH content was detected by the GSH assay kit.

**Intracellular O$_2$ generation**. The O$_2$ probe, [Ru(dpp)$_3$]Cl$_2$ (RDPP), was utilized to test the intracellular generation of O$_2$ via CLSM imaging. Briefly, MCF7 cells were seeded into 96-well plates for 24 h (37 °C, 5% CO$_2$). Then, the cells were treated as following: (1) FeOCl NSs, (2) FeOCl/FeOOH NSs, (3) FeOCl NSs + US, and (4) FeOCl/FeOOH NSs + US. The final concentration of NSs was 0.1 mg/mL. The condition for US treatment is 1 MHz, 1 W cm$^{-2}$, 50% duty cycle, 5 min. After that, RDPP (1 μM) was added into the above treated cells. After 24 h incubation, the treated MCF7 cells were washed three times by PBS. Finally, the intracellular O$_2$ concentration was determined by CLSM.

**Intracellular ROS generation**. The ROS probe, DCFH-DA, was utilized to test the intracellular generation of ROS via CLSM imaging. Briefly, MCF7 cells were seeded into 96-well plates for 24 h (37 °C, 5% CO$_2$). Then, the cells were treated as following: (1) FeOCl NSs, (2) FeOCl/FeOOH NSs, (3) FeOCl NSs + US, (4) FeOCl/FeOOH NSs + US, and (5) FeOCl NSs + US with H$_2$O$_2$. The final concentration of NSs was 0.1 mg/mL. The condition for US treatment is 1 MHz, 1 W cm$^{-2}$, 50% duty cycle, 5 min. The final added concentration of H$_2$O$_2$ in treatment 5 were 0.05 mM. After that, DCFH-DA (0.2 μM) was added into the above treated cells. After 1 h incubation, the treated MCF7 cells were washed three times by PBS. Finally, the intracellular ROS concentration was determined by CLSM.

**Anti-tumor therapy in vitro**. MCF7 and PC3 cells were incubated into 96-well plates for 24 h (37 °C, 5% $CO_2$). Subsequently, the old culture medium was replaced with the fresh one and treated as following groups: (1) FeOCl NSs, (2) FeOCl/FeOOH NSs, (3) FeOCl NSs + US, (4) FeOCl/FeOOH NSs + US, and (5) FeOCl NSs + US with $H_2O_2$. The final concentration of NSs was 0.1 mg/mL. The condition for US treatment is 1 MHz, 1 W cm$^{-2}$, 50% duty cycle, 5 min. The final added concentration of $H_2O_2$ in treatment 5 were 0.05 mM. The US treatment was carried out after NSs treated 12 h and removed from plates by PBS washing for three times. Finally, cell viabilities were determined by MTT assay.

**Xenograft tumor model**. All animal experiments were conducted according to the Guidelines for the Care and Use of Laboratory Animals of Tianjin University and were approved by the Animal Ethics Committee of the Tianjin University Laboratory Animal Center (Tianjin, China). The maximal tumor size permitted by Animal Ethics Committee of the Tianjin University Laboratory Animal Center is 2000 mm$^3$. The MCF7 tumor models were established by subcutaneous cell injection ($2 \times 10^6$ cells in 100 μL serum-free cell medium) into the Balb/c Nude mice (female, 6 weeks, 14–16 g). When the size of the tumors reached about 100 mm$^3$, the mice were randomly divided into different groups for different treatments.

**Pharmacokinetic study**. For the pharmacokinetics of FeOCl/FeOOH NSs in vivo, 200 μL of Cy7-PEG-NH$_2$ modified FeOCl/FeOOH NSs was i.v. injected in healthy C57BL/6 mice (female, 7 weeks, 16–18 g) with the dose of FeOCl/FeOOH NSs was 5 mg/kg. Afterwards, with different time interval, 20 μL blood was taken from the mice. Then, the fluorescence intensity of Cy7-PEG-NH$_2$ modified FeOCl/FeOOH NSs in blood was detected via BioTekmicroplate reader.

**Fluorescence imaging and biodistribution study in vivo**. FeOCl/FeOOH NSs with Cy7-PEG-NH$_2$ was injected intravenously into MCF7 tumor-bearing mice (female, 7 weeks, 16–18 g) via tail vein. The fluorescence was detected by Maestro2 In-Vivo Imaging System at different time post-injection. Subsequently, major organs and tumors of mice were obtained and imaged after the mice were killed by cervical dislocation. Fluorescence intensity of Cy7 was measured by Image-J. The intensity values were then divided by the weight (g) of each organ. For a more accurate quantitative measurement of the biodistribution, the FeOCl/FeOOH NSs, FeOCl/FeOOH NSs (10 mg kg$^{-1}$) was i.v. injected into MCF7 tumor-bearing mice. After 24 h post-injection, the mice were sacrificed and major organs and tumor were collected. The amount of extra Fe was measured by inductively coupled plasma-atomic emission spectrometry (ICP-AES).

**Anti-tumor therapy in vivo**. The MCF7 tumor-bearing mice (female, 7 weeks, 16–18 g) were randomly divided into six treatment groups with five mice for each group as follow: (1) PBS, (2) US, (3) FeOCl NSs, (4) FeOCl/FeOOH NSs, (5) FeOCl NSs + US and (6) FeOCl/FeOOH NSs + US. The injection dose of NSs was 5 mg/kg. The condition for US treatment is 1 MHz, 1 W cm$^{-2}$, 50% duty cycle. The exposure time of treatment with the US was 10 min. For group 5 and 6, the mice were exposed to the US successively at 24 h post-injection. Tumor size and body weight of each group were measured by a digital scale and caliper every 2 day for totally 14 days during the treatment. The inhibition rate was calculated by the following formula: "Tumor growth inhibition ratio = (Tumor volume of mice in control group-Tumor volume of mice in the treatment group)/Tumor volume of mice in control group × 100%".

**Biosafety in vivo**. To conduct the biosafety experiment, healthy C57BL/6 mice (female, 7 weeks, 16–18 g) were intravenously injected with FeOCl/FeOOH NSs in PBS (10 mg/mL). 24 h after injection, the representative cytokines including interleukin 6 (IL-6), tumor necrosis factor-α (TNF-α), and interferon-γ (IFN-γ) were measured by ELISA according to the manufacturer's instructions. Subsequently, the relative indexes in blood including albumin (ALB), aminotransferase (ALT), alanine aspartate aminotransferase (AST), blood urea nitrogen (BUN), creatinine (Cr), total protein (TP), C-reactive protein (CRP), lactate dehydrogenase (LDH), amylase (AMY), creatine kinase (CK), and γ-glutamyl transpeptidase (γ-GT), Red blood cells (RBC), white blood cells (WBC), hematocrit (HCT), hemoglobin (HGB), mean corpuscular hemoglobin (MCH), mean corpuscular hemoglobin concentration (MCHC), mean corpuscular volume (MCV), and platelet (PLT) counts were measured to further evaluate the biocompatibility and immune response of FeOCl/FeOOH NSs. After 1 month of treatment, the main organs were obtained for analysis by hematoxylin and eosin (H&E) staining.

**Reporting summary**. Further information on research design is available in the Nature Research Reporting Summary linked to this article.

## Data availability

The authors declare that all data supporting the findings of this study are available within the article and the Supplementary Information. Source data are provided with this paper.

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

## Acknowledgements

This study was financially supported by a grant from the National Natural Science Foundation of China (Grant No. 32071322), National Natural Science Funds for Excellent Young Scholar (Grant No. 32122044) and the Technology & Innovation Commission of Shenzhen Municipality (Grant No. JCYJ20210324113004010). All relevant funding was awarded to X.J.

## Author contributions

X.J. designed and supervised the project. X.J. designed the experimental strategies. X.J., Y.K., Z.M., Y.W., and C.P. performed the experiments and analyzed the data. X.J. and Y.K. wrote the manuscript with contributions from M.O., W.Z., and H.Z..

## Competing interests

The authors declare no competing interests.
