## [Peer Review File · Nature Communications]

Reviewers' Comments:

Reviewer #1:

Remarks to the Author:

In this manuscript, the authors developed a two-dimensional (2D) interplanar heterojunction (FeOCl/FeOOH NSs) for self-powered catalytic therapy via in situ $\cdot\text{OH}$ generation under ultrasound (US) irradiation. The self-supplying H_2O_2 capability guarantees abundant $\cdot\text{OH}$ production via the Fenton-like reaction catalyzed by both FeOCl and FeOOH, which displayed favorable therapeutic efficacy both in vitro and in vivo.

The design is interesting and the studies are well performed. Systematic in vitro and in vivo evaluations have demonstrated the therapeutic outcome of this nanoplatfrom-enabled synergistic chemodynamic therapy and sonodynamic therapy for completely eradicating the tumors without further recurrence. The overall study is well organized and performed. Thus I support acceptance of this work after addressing following revisions:

- 1) Is FeOCl/FeOOH NSs biodegradable in vivo? The biodegradability and long-term biosafety of FeOCl/FeOOH NSs should be investigated.
- 2) Is there any study regarding the long-term stability and/or behavior of the nanoplatfrom FeOCl/FeOOH NSs in mouse serum, PBS, and culture medium? This property is important for the in vivo use of these nanomedicines.
- 3) A calcein acetoxymethyl ester (calcein-AM)/propidium iodide (PI) staining assay is suggested to be performed to intuitively observe live and dead cells after various treatments.
- 4) The accumulation efficiency of FeOCl/FeOOH NSs in tumor tissue and main organs should be precisely quantified to better investigate their in vivo antitumor performance.
- 5) The tumor inhibition rates in different treatment groups should be quantified to better assess the in vivo therapeutic efficacy of FeOCl/FeOOH NSs under US irradiation.
- 6) Statistical analysis should be included for the in vitro cell investigation and in vivo tumor study data.

Reviewer #2:

Remarks to the Author:

This work reports a novel built-in electric field and Schottky barrier facilitated self-powered catalytic therapy based on 2D FeOCl/FeOOH interplanar heterojunction. The authors applied a facile wet-chemical treatment of FeOCl to bring so many surprises, such as constructing a promising interplanar heterojunction structure and facilitating ultrathin 2D nanosheets exfoliation. This exfoliated ultrathin interplanar heterojunction continuously and endlessly generates $\cdot\text{OH}$ through catalyzing an interesting cascade reaction using H_2O as the only substrate under US irradiation, which completely get rid of the limitation of tumor microenvironment. This strategy is very innovative. The authors have done an impressive job in comprehensively designing the self-powered catalytic therapy systems. Also, the manuscript, encompassing a considerable amount of data and analysis (in vitro, in vivo, materials characterization etc.), is well written and presented nicely. However, there are some minor aspects where the authors should further improve their study before consideration of publication.

- (1) Some important detailed analyses are missing in this manuscript. For characterizations of FeOCl/FeOOH heterojunction, a careful analysis of XRD and Raman spectra should be done. For the anti-tumor experiment in vitro, a detailed and quantitative analysis of ROS generation also should be added. The GSH consumption mechanism in FeOCl/FeOOH heterojunction need more fundamental insights.
- (2) The mechanism of PEGylation between PEG-NH₂ and FeOCl/FeOOH heterojunction is unclear. How do they conjugate each other? The information of zeta potential, molecular weight, purity, and production company etc. should be provided. How much PEG could be loaded on a nanosheets too? In addition, as the ROS generation is enabled by surface charge transfer processes, does the surface PEG-NH₂ conjugation block or suppress such catalytic processes? A detailed explanation or experiment of the influence of PEG-NH₂ conjugation is lacking.
- (4) The safety of nanomedicine or therapy is vital. Systemic exclusion and degradation of materials had to be carefully investigated. Furthermore, although blood tests of the prepared material have been tested, other biochemicals (CRP, LDH, AMY, CK, and γ -GT) and blood cells

(WBC, RBC, HGB, HCT, MCV, MCH, MCHC, and PLT) had to be assessed because ALB, ALT, AST, BUN, Cr, and TP are basically stable biochemical parameters whatever are intravenously injected. Normal cell lines had to be also applied for investigation of in vitro biocompatibility of materials. (5) Please carefully check all the figures in both the manuscript and supporting information. The captions of figures should be as detailed as possible, such as basic experimental or testing conditions should be added in their captions. And there were some typographical errors, such as the numbering order of the picture in Fig. 2. (6) Statistical analysis with statistical significance should be provided in some important cell and animal experiments.

Reviewer #3:

Remarks to the Author:

In the manuscript entitled Two Dimensional Interplanar Heterojunction Mediated Self Powered Catalytic Therapy with H₂O as the Only Substrate authors have reported a strategy for creating an interplanar heterojunction based on 2D FeOCl/FeOOH NSs capable of self-supplying H₂O₂ without limitation of TME and a powerful ability for tumor inhibition using a wet chemical method based on alkali replacement reaction. After PEGylation, the FeOCl/FeOOH NSs functionalized with a characteristic interplanar heterojunction for efficient H₂O₂ self-supplying applying H₂O as the only substrate was developed and exhibited a highly efficient chemo dynamic effect using US irradiation as claimed by the authors. This manuscript is comprehensive and may be of interest to wide readership. However, the manuscript has several substantive inconsistencies and major problems that need to be addressed prior to a final editorial decision.

1. The term self-powered catalytic therapy is not suitable as the catalytic activity is mostly triggered by ultrasound irradiation. Hence, this term should be justified in more appropriate way as this is not thermo responsive therapy triggered by the body heat which could be called self-powered.
2. The overall manuscript has no description of piezo catalytic activity, if the activity is US driven, how is it different from piezo catalytic therapy?
3. If the activity is piezo catalytic, which seems it is, the authors should demonstrate the piezo response of the material such as PFM study.
4. The manuscript is entirely written highlighting the material point of view and there is negligible amount of importance mentioned from the tumor point of view. The authors should clearly describe the motivation in the story as to why this study is important for treating tumors and why the particular target?
5. The authors also need to mention how and why this reported work is beneficial and advantageous over other catalytic techniques.
6. The captions in the figure 2 are repetitive and very confusing for example f,g, and h are captioned twice and it is confusing in the results and discussion section what the authors are trying to convey. These figures should be re captioned and the discussion should be re-written accordingly.
7. In figure 3, XRD indexing is missing and even in the description section, the phases are not mentioned which is not clear for readers. Also, the same for Raman spectroscopy this should be mentioned and explained in detail.
8. Since PEGylation of the NSs is done for the final study, all the materials characterizations should be repeated after PEGylation and compared to non PEGylation samples as to prove that the materials property has not been affected. This is an important point, which the authors must address.
9. Supplementary figure 4 should be clearly captioned and addressed in the manuscript clearly as well.
10. The authors have focused on the self-supplying H₂O₂ ability of the material. However, the authors need to provide the H₂O₂ as well as .OH production data of their material in water with and without ultrasound. The data should be provided with and without PEGylation of the materials. Authors may refer to (Nature Communications volume 12, Article number: 180 (2021) and Nano Energy 57 (2019) 14–2115 for these experiments.
11. In figure 4c,d no generation of in H₂O₂ FeOCl is due to unfavorable band position of FeOCl to generate H₂O₂ as shown in figure 5g. Authors should rectify this description in terms of band position.

12. The position for .OH production described in the schematic of figure 5g is incorrect and should be re-checked.

13. The authors should also check if the hydroxyl radical generation is the result of band tailoring or fenton reactions using the generated H₂O₂. Authors may confirm this by exactly evaluating the favorable bond position for the ROS generation.

14. For in vitro experiments, amount of H₂O₂ used was significantly high and for in vivo experiments no externally supplied H₂O₂ was used. The amount of H₂O₂ in the TME may not be sufficiently high and in this case, what will be the anti-tumor performance of the material. Was H₂O₂ also supplied along with the material in the in vivo experiments?

15. In most of the experiments H₂O₂ was externally supplied, thus H₂O as the only substrate may not be a reasonable claim. Authors should rethink about this claim with proper justification.

16. It will be an interesting study as how this therapy can be applied on diabetic subjects. What according to the authors will be a challenge or will the catalytic performance remain unchanged irrespective of the subject's health conditions?

Dear editors and reviewers,

We sincerely appreciate your thoughtful and professional comments on our manuscript. We have adopted all of the suggestions and revised the manuscript accordingly. We hope that the revised manuscript has been improved to the level of the reviewers' satisfaction. We also hope that you will now find it suitable for publication in *Nature Communications*.

Reviewer #1 (Remarks to the Author):

In this manuscript, the authors developed a two-dimensional (2D) interplanar heterojunction (FeOCl/FeOOH NSs) for self-powered catalytic therapy via in situ $\cdot\text{OH}$ generation under ultrasound (US) irradiation. The self-supplying H_2O_2 capability guarantees abundant $\cdot\text{OH}$ production via the Fenton-like reaction catalyzed by both FeOCl and FeOOH, which displayed favorable therapeutic efficacy both in vitro and in vivo. The design is interesting and the studies are well performed. Systematic in vitro and in vivo evaluations have demonstrated the therapeutic outcome of this nanoplatform-enabled synergistic chemodynamic therapy and sonodynamic therapy for completely eradicating the tumors without further recurrence. The overall study is well organized and performed. Thus I support acceptance of this work after addressing following revisions:

Response: Thanks for the reviewer's positive comments. We have followed the reviewer's comments and performed additional experiments to address the points raised by the reviewer. Please see below point-by-point responses.

1) Is FeOCl/FeOOH NSs biodegradable in vivo? The biodegradability and long-term biosafety of FeOCl/FeOOH NSs should be investigated.

Response: We very much appreciate the reviewer's thoughtful and helpful comments. We also very agree with the reviewer. The biodegradability, especially in PBS, mouse serum, and medium, and long-term biosafety of FeOCl/FeOOH NSs have been investigated in details and added in our revised manuscript.

(1) Biodegradability

“Having demonstrated the effective tumor eradication induced by the 2D interplanar heterojunction FeOCl/FeOOH NSs, their biodegradability by dialysis in PBS, medium, and serum solutions were further investigated. The UV-vis-NIR absorbance of the residual FeOCl/FeOOH NSs left in the dialysis bag was detected every 2 days during the 14 days dialysis to evaluate the degradation rate. As shown in Fig. 10a-10c, very slow biodegradation behaviors of FeOCl/FeOOH NSs were observed in PBS and medium solutions, demonstrating a good storage stability. A relatively fast biodegradability behavior of FeOCl/FeOOH NSs was exhibited in serum, which illustrated a desired biodegradability *in vivo*. To further reveal the biodegradation mechanism of FeOCl/FeOOH NSs, TEM images and size distributions of FeOCl/FeOOH NSs after 14 days in different solutions were detected, and showed in Fig. 10d and 10e. A kind of degradation phenomenon spreading from edge to center was observed in their TEM images. The gradually decreased size distribution also demonstrated the biodegradability of FeOCl/FeOOH NSs. Moreover, the

biodegradability of FeOCl/FeOOH NSs *in vivo* were further analyzed by ICP. As shown in Fig. 10f, the accumulated FeOCl/FeOOH NSs in normal organs and tissues could be gradually degraded and excreted by the body over time.”

(2) Biosafety

“The biocompatibility of FeOCl/FeOOH NSs was further confirmed through hematological, and immunological results. For immune analysis, the serum level of IL-6, TNF- α , and IFN- γ at 2 h and 24 h post intravenous injection of FeOCl/FeOOH NSs was measured. As exhibited in Supplementary Fig. 22, all cytokine levels from the FeOCl/FeOOH NSs treated group showed no statistically significant differences compared with the control group, confirming that the prepared FeOCl/FeOOH NSs have good biocompatibility and biosafety *in vivo*. Besides, there were no obvious differences of biochemical detection detected between the mice of control group and that administrated with FeOCl/FeOOH NSs for 1, 7, and 14 days in the blood parameters of albumin (ALB), aminotransferase (ALT), alanine aspartate aminotransferase (AST), blood urea nitrogen (BUN), creatinine (Cr), total protein (TP), C-reactive protein (CRP), lactate dehydrogenase (LDH), amylase (AMY), creatine kinase (CK), and γ -glutamyl transpeptidase (γ -GT), (Fig. 10g). Moreover, the hematological detection was further carried out for evaluating the biocompatibility of FeOCl/FeOOH NSs. Red blood cells (RBC), white blood cells (WBC), hematocrit (HCT), hemoglobin (HGB), mean corpuscular hemoglobin (MCH), mean corpuscular hemoglobin concentration (MCHC), mean corpuscular volume (MCV), and platelet (PLT) counts were measured (Fig. 10h). Compared with the control group, although there were a little increase of some indicators, no statistically significant difference of the FeOCl/FeOOH NSs-treated groups with PBS-treated groups in all the parameters after i.v. injection for 7 and 14 days. Supported by the above results, FeOCl/FeOOH NSs were regarded as relatively biocompatible and biosafe *in vivo*.”

Fig. 10 Biodegradation behavior of FeOCl/FeOOH NSs. UV-vis-NIR absorbance spectra of layered FeOCl/FeOOH NSs co-incubated with **a** PBS, **b** Serum and **c**

Medium solution for 14 days. **d** TEM images of layered FeOCl/FeOOH NSs and FeOCl/FeOOH NSs co-incubated with PBS, serum, and medium solution for 14 days, scale bar = 50 nm. **e** Size distribution of layered FeOCl/FeOOH NSs and FeOCl/FeOOH NSs co-incubated with PBS, serum, and medium solution for 14 days. **f** Biodistribution of FeOCl/FeOOH NSs in tumor-bearing mice by ICP measurement. The data show mean \pm s.d., $n = 3$ biologically independent mice. **g** Blood hematology analysis of Balb/c mice treated with PBS or FeOCl/FeOOH NSs. **h** Blood biochemistry analysis of Balb/c mice treated with PBS or FeOCl/FeOOH NSs.

Supplementary Figure 22. Serum levels of IL-6, IFN- γ , TNF- α , and IL-12+P40 in mice at 2 and 24 h post i.v. injection of PBS versus FeOCl/FeOOH NSs. The data show mean \pm s.d., $n = 3$ biologically independent mice.

2) Is there any study regarding the long-term stability and/or behavior of the nanopatform FeOCl/FeOOH NSs in mouse serum, PBS, and culture medium? This property is important for the in vivo use of these nanomedicines.

Response: Thank you for this valuable comment. The biodegradable behaviors of the nanopatform FeOCl/FeOOH NSs in mouse serum, PBS, and culture medium have been investigated and added in our revised manuscript, and the corresponding explanations and analysis were presented in question 1.

3) A calcein acetoxymethyl ester (calcein-AM)/propidium iodide (PI) staining assay is suggested to be performed to intuitively observe live and dead cells after various treatments.

Response: We appreciate this helpful comment. The related results and discussion are added in the revised manuscript as follows:

“A calcein acetoxymethyl ester (calcein-AM)/propidium iodide (PI) staining assay was performed to intuitively observe live and dead cells after various treatments. The results exhibited in Supplementary Fig. 14 showed obvious apoptosis in FeOCl NSs

group, FeOCl/FeOOH NSs group and FeOCl NSs + US group, while the maximum amount of apoptosis was observed in FeOCl/FeOOH NSs + US group.”

Supplementary Figure 14. Fluorescence images of MCF7 cells stained with Calcein-AM (live cells, green fluorescence) and PI (dead cells, red fluorescence) after treated with different conditions (scale bar = 150 μ m).

4) The accumulation efficiency of FeOCl/FeOOH NSs in tumor tissue and main organs should be precisely quantified to better investigate their in vivo antitumor performance.

Response: Thanks for this valuable comment. The accumulation efficiency of FeOCl/FeOOH NSs in tumor tissue and main organs has been shown in the Fig 8d. The more detailed and in-depth analysis have been described in the manuscript:

“Moreover, an inductively coupled plasma emission spectrometer (ICP) was applied to more precisely quantified the accumulation efficiency of FeOCl/FeOOH NSs in tumor tissue and main organs. As shown in Fig. 8d, after 24 h post-injection, a high tumor accumulation efficiency of 16.5% was detected. The high accumulation of drugs in the tumor site is the cornerstone of excellent antitumor performance. Meanwhile, the accumulation efficiency of FeOCl/FeOOH NSs in liver reached to 31%, which was caused by the absorption of the mononuclear phagocyte system (Fig. 8d).”

5) The tumor inhibition rates in different treatment groups should be quantified to better assess the in vivo therapeutic efficacy of FeOCl/FeOOH NSs under US irradiation.

Response: The reviewer’s comment is very constructive and useful. According to the results of animal experiments, the related discussion of the tumor inhibition rates was added in manuscript:

In the “Materials and Methods” section:

“The inhibition rate was calculated by the following formula: “Tumor growth inhibition ratio = (Tumor volume of mice in control group-Tumor volume of mice in treatment group)/Tumor volume of mice in control group × 100%””

In the “Results and Discussion” section:

“The tumor inhibition rate based on the average volume of tumors exhibited that the tumor inhibition rate of the Treatment 6 group was 99.8% (Supplementary Fig. 20).”

Supplementary Figure 20. Tumor growth inhibition ratio of each group.

6) Statistical analysis should be included for the in vitro cell investigation and in vivo tumor study data.

Response: We very much appreciate the reviewer’s thoughtful and helpful comments. The statistical analysis with statistical significance has been added in our important cell and animal experiments.

Fig. 6 Biocompatibility and cytotoxicity of FeOCl NSs and FeOCl/FeOOH NSs. a Relative viability of normal human cells (HL-7702 and HEK293) after incubation with FeOCl NSs or FeOCl/FeOOH NSs for 24 h. **b** Relative viability of MCF7 cells after incubation with FeOCl NSs or FeOCl/FeOOH NSs under different treatments for 24 h. **c** The generation of O₂ of MCF7 cells after incubation with FeOCl NSs or FeOCl/FeOOH NSs under different treatments. **d** The degradation of GSH of MCF7 cells after incubation with FeOCl NSs or FeOCl/FeOOH NSs under different treatments. **e** The consumption of H₂O₂ of MCF7 cells after incubation with FeOCl NSs or FeOCl/FeOOH NSs under different treatments. **f** FCM images of MCF7 cells after incubation with FeOCl NSs and FeOCl/FeOOH NSs under different treatments for 12 h. The condition for US treatment is 1 MHz, 1 W cm⁻², 50% duty cycle, 5 min.

Fig. 7 Intracellular ROS generation, O₂ generation, and DNA damage of MCF7 cells under different treatments. **a** The intercellular ROS generation and **c** the O₂ generation of MCF7 cells under different treatments with FeOCl NSs or FeOCl/FeOOH NSs, scale bar = 100 μ m. **b** and **d** Fluorescence quantitative analysis of intracellular ROS and O₂ generation. **e** Intracellular ROS generation detected by FCM. **f** Representative confocal microscopy images of the MCF7 cells (scale bars = 30 μ m) after different treatments. The nuclei were stained by DAPI (blue), and the γ H2AX foci per nucleus were stained by anti- γ H2AX antibody (red). **g** Quantification of the percentages of cells with >10 γ H2AX foci numbers by confocal microscopy. **h** *In vitro* DNA damage of the cells after different treatments were measured by

8-OHdG assay. The condition for US treatment is 1 MHz, 1 W cm⁻², 50% duty cycle, 10 min.

Fig. 8 *In vivo* imaging and anti-tumor performance of FeOCl NSs and FeOCl/FeOOH NSs. **a** Schematic diagram of treatment. **b** *In vivo* fluorescence images of nude mice after i.v. administration of NSs, and the ex vivo fluorescence images of the tumor and major organs at 24 h post-injection of Cy 7 loaded FeOCl/FeOOH NSs. **c** Blood circulation of FeOCl/FeOOH NSs. **d** Bio-distribution of FeOCl/FeOOH NSs under different time. **e** Tumor growth curves of MCF7 tumor-bearing nude mice. **f** Representative tumor photos in different groups after 14 days of treatment. **g** Tumor growth curves of MCF7 tumor-bearing nude mice. **h** Survival rate of mice undergoing different treatments. **i** Body weight of mice during treatment. The condition for US treatment is 1 MHz, 1 W cm⁻², 50% duty cycle, 10 min.

Reviewer #2 (Remarks to the Author):

This work reports a novel built-in electric field and Schottky barrier facilitated self-powered catalytic therapy based on 2D FeOCl/FeOOH interplanar heterojunction. The authors applied a facile wet-chemical treatment of FeOCl to bring so many surprises, such as constructing a promising interplanar heterojunction structure and facilitating ultrathin 2D nanosheets exfoliation. This exfoliated ultrathin interplanar heterojunction continuously and endlessly generates $\cdot\text{OH}$ through catalyzing an interesting cascade reaction using H_2O as the only substrate under US irradiation, which completely get rid of the limitation of tumor microenvironment. This strategy is very innovative. The authors have done an impressive job in comprehensively designing the self-powered catalytic therapy systems. Also, the manuscript, encompassing a considerable amount of data and analysis (in vitro, in vivo, materials characterization etc.), is well written and presented nicely. However, there are some minor aspects where the authors should further improve their study before consideration of publication.

Response: Thanks for the reviewer's positive comments. We have followed the reviewer's comments and performed additional experiments to address the points raised by the reviewer. Please see below point-by-point responses.

(1) Some important detailed analyses are missing in this manuscript. For characterizations of FeOCl/FeOOH heterojunction, a careful analysis of XRD and Raman spectra should be done. For the anti-tumor experiment in vitro, a detailed and quantitative analysis of ROS generation also should be added. The GSH consumption mechanism in FeOCl/FeOOH heterojunction need more fundamental insights.

Response: We appreciate this helpful comment. These important detailed analyses were added in our revised manuscript.

The characterizations of FeOCl/FeOOH heterojunction including XRD and Raman analysis:

“In the XRD spectra of FeOCl NSs (Fig. 3b), all the XRD peaks are well-matched with JCPDS card No. 01-0081 corresponding to orthorhombic structured FeOCl nanocrystals, which demonstrated the high purity of synthesized FeOCl. After alkali etching, another respective crystal structures were observed, which corresponded with FeOOH (JCPDS No. 01-0136), illustrating the successfully edge decoration. In the Raman spectra (Fig. 3c), the peaks of FeOCl at A_{1g} mode (212 cm^{-1}) and E_g mode (291 cm^{-1}) are related to Fe–O stretching vibration. After alkali etching, the characteristic peaks of FeOOH at 240 cm^{-1} and 400 cm^{-1} were all exhibited in the spectrum of FeOCl/FeOOH NSs, which further demonstrated the successfully edge decoration.”

Fig. 3 Chemical composition and structure characterization of layered FeOCl NSs and FeOCl/FeOOH NSs. **a** XPS spectra of FeOCl NSs and FeOCl/FeOOH NSs. **b** XRD spectra of FeOCl NSs and FeOCl/FeOOH NSs. **c** Raman shift spectra of FeOCl NSs and FeOCl/FeOOH NSs. **d** and **g** HRXPS spectra of Fe 2p in FeOCl NSs and FeOCl/FeOOH NSs. **e** and **h** HRXPS spectra of Cl 2p in FeOCl NSs and FeOCl/FeOOH NSs. **f** and **i** HRXPS spectra of O 1s in FeOCl NSs and FeOCl/FeOOH NSs.

The detailed and quantitative analysis of ROS generation:

“As shown in Fig. 7a and 7b, cancer cells treated with FeOCl NSs and FeOCl/FeOOH NSs alone showed a slightly $\cdot\text{OH}$ production in contrast to control group, meaning FeOCl NSs and FeOCl/FeOOH NSs induced the weak Fenton reaction with intrinsic H_2O_2 in TME. FeOCl/FeOOH NSs plus US irradiation treatment triggered the strongest green fluorescence signal, due to the self-supplying H_2O_2 ability of interplanar heterojunction. The difference between FeOCl NSs + US group and FeOCl NSs + US + H_2O_2 group further demonstrated the significant role of H_2O_2 to CDT and the excellent self-supplying H_2O_2 ability of this interplanar heterojunction. The flow cytometry detection was also applied to test the intracellular ROS content after different treatments. Supplementary Fig. 16 and Fig. 7e exhibited a similar ROS content with that of confocal images.”

Fig. 7 Intracellular ROS generation, O₂ generation, and DNA damage of MCF7 cells under different treatments. **a** The intercellular ROS generation and **c** the O₂ generation of MCF7 cells under different treatments with FeOCl NSs or FeOCl/FeOOH NSs, scale bar = 100 μm. **b** and **d** Fluorescence quantitative analysis of intracellular ROS and O₂ generation. **e** Intracellular ROS generation detected by FCM. **f** Representative confocal microscopy images of the MCF7 cells (scale bars = 30 mm) after different treatments. The nuclei were stained by DAPI (blue), and the γH2AX foci per nucleus were stained by anti-γH2AX antibody (red). **g** Quantification

of the percentages of cells with >10 γ H2AX foci numbers by confocal microscopy. **h** *In vitro* DNA damage of the cells after different treatments were measured by 8-OHdG assay. The condition for US treatment is 1 MHz, 1 W cm^{-2} , 50% duty cycle, 10 min.

The GSH consumption mechanism in FeOCl/FeOOH heterojunction:

“GSH, one of most important keys to regulate intracellular REDOX equilibrium, was over-expressed in tumor cells. The intratumoral over-expressed GSH (0.32 eV oxidation potential of GSH/GSSG) was also oxidized by other oxidants, such as $\text{Fe}^{2+}/\text{Fe}^{3+}$ pairs and separated holes. As shown in Fig. 4e-4f and Supplementary Fig. 10, for FeOCl NSs and FeOCl/FeOOH NSs groups, the ionization of $\text{Fe}^{2+}/\text{Fe}^{3+}$ could be the primary reason for the oxidation of GSH. For FeOCl NSs + US and FeOCl/FeOOH NSs + US groups, the excited holes with strong oxidated ability contributed to the other part of oxidation of GSH. Due the Fenton reaction recirculates Fe^{3+} , FeOCl NSs + US and FeOCl/FeOOH NSs + US groups with extra added H_2O_2 also accelerated the consumption of GSH.”

(2) The mechanism of PEGylation between PEG-NH₂ and FeOCl/FeOOH heterojunction is unclear. How do they conjugate each other? The information of zeta potential, molecular weight, purity, and production company etc. should be provided. How much PEG could be loaded on a nanosheets too? In addition, as the ROS generation is enabled by surface charge transfer processes, does the surface PEG-NH₂ conjugation block or suppress such catalytic processes? A detailed explanation or experiment of the influence of PEG-NH₂ conjugation is lacking.

Response: Thanks for this comment. Due to the negatively charged surface of prepared FeOCl/FeOOH heterojunction, the positively charged PEG(5k)-NH₂ could be easily absorbed on their surfaces *via* electrostatic attraction. In our revised manuscript, the PEGylation process, Zeta potentials of FeOCl/FeOOH heterojunction, and the amount of absorbed PEG-NH₂ were added.

Experimental section:

“**PEGylation of FeOCl NSs and FeOCl/FeOOH NSs.** In order to improve the stability and biocompatibility of FeOCl NSs and FeOCl/FeOOH NSs, NSs were further modified with PEG(5k)-NH₂. First, PEG(5k)-NH₂ (10 mg) was added into NSs suspension and then the suspension was ultrasonicated for 30 min. After stirred for 12 h, the mixture was washed three times by centrifugation at 2500 rpm (4°C) for 30 min. Afterwards, the PEGylated NSs were resuspended in PBS stored at 4°C for further use. To prepare Cy7-labeled FeOCl NSs and FeOCl/FeOOH NSs, The Cy7-labeled PEG(5k)-NH₂ (Cy7-PEG(5k)-NH₂) and PEG(5k)-NH₂ (1:4) were then coated on the surface of NSs to generate Cy7-labeled NSs using the aforementioned method.”

Results section:

“Due to the biocompatibility and dispersibility are crucial for biomedical applications, the as-prepared FeOCl NSs and FeOCl/FeOOH NSs were modified by PEG-NH₂. PEG(5k)-NH₂ positively charged was absorbed on the negatively charged surface of

prepared FeOCl NSs and FeOCl/FeOOH NSs (Supplementary Fig. 3) *via* electrostatic attraction. The UV-vis-NIR absorbance spectra exhibited that there was no significant difference before and after PEG modification (Supplementary Fig. 4). The amount of PEG(5k)-NH₂ that was coated on the surface of the FeOCl/FeOOH NSs was $\approx 22.4\%$ (w/w) as measured by thermo gravimetric analysis (TGA).”

Supplementary Figure 3. The Zeta potentials of prepared FeOCl NSs and FeOCl/FeOOH NSs.

In addition, analysis of the effect of surface modification on catalytic activity has been added in our revised manuscript.

“PEGylation of FeOCl NSs and FeOCl/FeOOH NSs possess an enhanced dispersion in water, phosphate buffer saline (PBS), and cell culture medium compared with bare NSs (Supplementary Fig. 5). So that, more catalytic active sites of PEGylated nanocatalyst were exposed to substrates than that of aggregated nanocatalysts. In addition, since the catalytic substrates are dissolved in water, PEG modification can effectively increase the surface hydrophilicity of nanomaterials, and thus increase the adsorption of substrate molecules, such as dissolved O₂, GSH, and H₂O, on the surface of prepared NSs.”

(4) The safety of nanomedicine or therapy is vital. Systemic exclusion and degradation of materials had to be carefully investigated. Furthermore, although blood tests of the prepared material have been tested, other biochemicals (CRP, LDH, AMY, CK, and γ -GT) and blood cells (WBC, RBC, HGB, HCT, MCV, MCH, MCHC, and PLT) had to be assessed because ALB, ALT, AST, BUN, Cr, and TP are basically stable biochemical parameters whatever are intravenously injected. Normal cell lines had to be also applied for investigation of *in vitro* biocompatibility of materials.

Response: Thanks for this professional comment. The systemic exclusion and degradation and the safety of this nanomedicine was analyzed in more details.

“Biodegradability and biosafety. Having demonstrated the effective tumor eradication induced by the 2D interplanar heterojunction FeOCl/FeOOH NSs, their biodegradability by dialysis in PBS, medium, and serum solutions were further investigated. The UV-vis-NIR absorbance of the residual FeOCl/FeOOH NSs left in the dialysis bag was detected every 2 days during the 14 days dialysis to evaluate the degradation rate. As shown in Fig. 10a-10c, very slow biodegradation behaviors of FeOCl/FeOOH NSs were observed in PBS and medium solutions, demonstrating a good storage stability. A relatively fast biodegradability behavior of FeOCl/FeOOH NSs was exhibited in serum, which illustrated a desired biodegradability *in vivo*. To further reveal the biodegradation mechanism of FeOCl/FeOOH NSs, TEM images and size distributions of FeOCl/FeOOH NSs after 14 days in different solutions were detected, and showed in Fig. 10d and 10e. A kind of degradation phenomenon spreading from edge to center was observed in their TEM images. The gradually decreased size distribution also demonstrated the biodegradability of FeOCl/FeOOH NSs. Moreover, the biodegradability of FeOCl/FeOOH NSs *in vivo* were further analyzed by ICP. As shown in Fig. 10f, the accumulated FeOCl/FeOOH NSs in normal organs and tissues could be gradually degraded and excreted by the body over time. As shown in Supplementary Fig. 21, no sign of normal organ injury was detected, demonstrating the excellent *in vivo* biosafety of the prepared FeOCl/FeOOH NSs. The biocompatibility of FeOCl/FeOOH NSs was further confirmed through hematological, and immunological results. For immune analysis, the serum level of IL-6, TNF- α , and IFN- γ at 2 h and 24 h post intravenous injection of FeOCl/FeOOH NSs was measured. As exhibited in Supplementary Fig. 22, all cytokine levels from the FeOCl/FeOOH NSs treated group showed no statistically significant differences compared with the control group, confirming that the prepared FeOCl/FeOOH NSs have good biocompatibility and biosafety *in vivo*. Besides, there were no obvious differences of biochemical detection detected between the mice of control group and that administrated with FeOCl/FeOOH NSs for 1, 7, and 14 days in the blood parameters of albumin (ALB), aminotransferase (ALT), alanine aspartate aminotransferase (AST), blood urea nitrogen (BUN), creatinine (Cr), total protein (TP), C-reactive protein (CRP), lactate dehydrogenase (LDH), amylase (AMY), creatine kinase (CK), and γ -glutamyl transpeptidase (γ -GT), (Fig. 10g). Moreover, the hematological detection was further carried out for evaluating the biocompatibility of FeOCl/FeOOH NSs. Red blood cells (RBC), white blood cells (WBC), hematocrit (HCT), hemoglobin (HGB), mean corpuscular hemoglobin (MCH), mean corpuscular hemoglobin concentration (MCHC), mean corpuscular volume (MCV), and platelet (PLT) counts were measured (Fig. 10h). Compared with the control group, although there were a little increase of some indicators, no statistically significant difference of the FeOCl/FeOOH NSs-treated groups with PBS-treated groups in all the parameters after i.v. injection for 7 and 14 days. Supported by the above results, FeOCl/FeOOH NSs were regarded as relatively biocompatible and biosafe *in vivo*.”

Fig. 10 Biodegradation behavior of FeOCl/FeOOH NSs. UV-vis-NIR absorbance spectra of layered FeOCl/FeOOH NSs co-incubated with **a** PBS, **b** Serum and **c**

Medium solution for 14 days. **d** TEM images of layered FeOCl/FeOOH NSs and FeOCl/FeOOH NSs co-incubated with PBS, serum, and medium solution for 14 days, scale bar = 50 nm. **e** Size distribution of layered FeOCl/FeOOH NSs and FeOCl/FeOOH NSs co-incubated with PBS, serum, and medium solution for 14 days. **f** Biodistribution of FeOCl/FeOOH NSs in tumor-bearing mice by ICP measurement. The data show mean \pm s.d., $n = 3$ biologically independent mice. **g** Blood hematology analysis of Balb/c mice treated with PBS or FeOCl/FeOOH NSs. **h** Blood biochemistry analysis of Balb/c mice treated with PBS or FeOCl/FeOOH NSs.

Supplementary Figure 21. HE images of major organs (heart, liver, spleen, lung, and kidney) after different treatments.

Supplementary Figure 22. Serum levels of IL-6, IFN- γ , TNF- α , and IL-12+P40 in mice at 2 and 24 h post i.v. injection of PBS versus FeOCl/FeOOH NSs. The data show mean \pm s.d., $n = 3$ biologically independent mice.

The biocompatibility of FeOCl/FeOOH NSs for normal cells:

“As exhibited in Fig. 6a, both the FeOCl NSs and FeOCl/FeOOH NSs exhibited good biocompatibility and safety to normal cells. By contrast, the FeOCl NSs and FeOCl/FeOOH NSs showed obviously specific cytotoxicity to cancer cells (Fig. 6b and Supplementary Fig. 13). The generated more cytotoxic \cdot OH *via* Fenton reaction resulted from high content of H₂O₂ in TME would be the primary cause for the specific cytotoxicity to cancer cells of FeOCl NSs and FeOCl/FeOOH NSs.”

Fig. 6 Biocompatibility and cytotoxicity of FeOCl NSs and FeOCl/FeOOH NSs. a Relative viability of normal human cells (HL-7702 and HEK293) after incubation with FeOCl NSs or FeOCl/FeOOH NSs for 24 h. **b** Relative viability of MCF7 cells after incubation with FeOCl NSs or FeOCl/FeOOH NSs under different treatments for 24 h. **c** The generation of O_2 of MCF7 cells after incubation with FeOCl NSs or FeOCl/FeOOH NSs under different treatments. **d** The degradation of GSH of MCF7 cells after incubation with FeOCl NSs or FeOCl/FeOOH NSs under different treatments. **e** The consumption of H_2O_2 of MCF7 cells after incubation with FeOCl NSs or FeOCl/FeOOH NSs under different treatments. **f** FCM images of MCF7 cells after incubation with FeOCl NSs and FeOCl/FeOOH NSs under different treatments for 12 h. The condition for US treatment is 1 MHz, $1 W cm^{-2}$, 50% duty cycle, 5 min.

(5) Please carefully check all the figures in both the manuscript and supporting information. The captions of figures should be as detailed as possible, such as basic experimental or testing conditions should be added in their captions. And there were some typographical errors, such as the numbering order of the picture in Fig. 2.

Response: Thanks for this valuable comment. We are very sorry for this negligence. The detailed description of captions has been added in our revised manuscript, and the typographical errors in Fig. 2 also revised.

Fig. 2 Morphology and composition characterization of layered FeOCl NSs and FeOCl/FeOOH NSs. SEM images of layered **a** FeOCl and **e** FeOCl/FeOOH, scale bar = 100 nm. Enlarged SEM images of layered **b** FeOCl and **f** FeOCl/FeOOH, scale bar = 15 nm. TEM images of ultrathin **c** FeOCl NSs and **g** FeOCl/FeOOH NSs, scale bar = 100 nm. HRTEM images of ultrathin **d** FeOCl NSs and **h** FeOCl/FeOOH NSs, scale bar = 3 nm. SEM-EDS mapping of **i** FeOCl and **j** FeOCl/FeOOH: red (Fe), green (O), and yellow (Cl), scale bar = 100 nm. **k** AFM images of FeOCl/FeOOH NSs, scale bar = 100 nm. **l** Thickness of FeOCl/FeOOH NSs. **m** 3D AFM images of FeOCl/FeOOH NSs, scale bar = 100 nm.

(6) Statistical analysis with statistical significance should be provided in some important cell and animal experiments.

Response: We very much appreciate the reviewer's thoughtful and helpful comments. The statistical analysis with statistical significance has been added in our important cell and animal experiments.

Fig. 6 Biocompatibility and cytotoxicity of FeOCI NSs and FeOCI/FeOOH NSs. **a** Relative viability of normal human cells (HL-7702 and HEK293) after incubation with FeOCI NSs or FeOCI/FeOOH NSs for 24 h. **b** Relative viability of MCF7 cells after incubation with FeOCI NSs or FeOCI/FeOOH NSs under different treatments for 24 h. **c** The generation of O_2 of MCF7 cells after incubation with FeOCI NSs or FeOCI/FeOOH NSs under different treatments. **d** The degradation of GSH of MCF7 cells after incubation with FeOCI NSs or FeOCI/FeOOH NSs under different treatments. **e** The consumption of H_2O_2 of MCF7 cells after incubation with FeOCI NSs or FeOCI/FeOOH NSs under different treatments. **f** FCM images of MCF7 cells after incubation with FeOCI NSs and FeOCI/FeOOH NSs under different treatments for 12 h. The condition for US treatment is 1 MHz, 1 W cm^{-2} , 50% duty cycle, 5 min.

Fig. 7 Intracellular ROS generation, O₂ generation, and DNA damage of MCF7 cells under different treatments. **a** The intercellular ROS generation and **c** the O₂ generation of MCF7 cells under different treatments with FeOCl NSs or FeOCl/FeOOH NSs, scale bar = 100 μm. **b** and **d** Fluorescence quantitative analysis of intracellular ROS and O₂ generation. **e** Intracellular ROS generation detected by FCM. **f** Representative confocal microscopy images of the MCF7 cells (scale bars = 30 μm) after different treatments. The nuclei were stained by DAPI (blue), and the γH2AX foci per nucleus were stained by anti-γH2AX antibody (red). **g** Quantification of the percentages of cells with >10 γH2AX foci numbers by confocal microscopy. **h** *In vitro* DNA damage of the cells after different treatments were measured by 8-OHdG assay. The condition for US treatment is 1 MHz, 1 W cm⁻², 50% duty cycle, 10 min.

Fig. 8 *In vivo* imaging and anti-tumor performance of FeOCl NSs and FeOCl/FeOOH NSs. **a** Schematic diagram of treatment. **b** *In vivo* fluorescence images of nude mice after i.v. administration of NSs, and the ex vivo fluorescence images of the tumor and major organs at 24 h post-injection of Cy7 loaded FeOCl/FeOOH NSs. **c** Blood circulation of FeOCl/FeOOH NSs. **d** Bio-distribution of FeOCl/FeOOH NSs under different time. **e** Tumor growth curves of MCF-7 tumor-bearing nude mice. **f** Representative tumor photos in different groups after 14 days of treatment. **g** Tumor growth curves of MCF7 tumor-bearing nude mice. **h** Survival rate of mice undergoing different treatments. **i** Body weight of mice during treatment. The condition for US treatment is 1 MHz, 1 W cm⁻², 50% duty cycle, 10 min.

Reviewer #3 (Remarks to the Author):

In the manuscript entitled Two Dimensional Interplanar Heterojunction Mediated Self Powered Catalytic Therapy with H₂O as the Only Substrate authors have reported a strategy for creating an interplanar heterojunction based on 2D FeOCl/FeOOH NSs capable of self-supplying H₂O₂ without limitation of TME and a powerful ability for tumor inhibition using a wet chemical method based on alkali replacement reaction. After PEGylation, the FeOCl/FeOOH NSs functionalized with a characteristic interplanar heterojunction for efficient H₂O₂ self-supplying applying H₂O as the only substrate was developed and exhibited a highly efficient chemo dynamic effect using US irradiation as claimed by the authors. This manuscript is comprehensive and may be of interest to wide readership. However, the manuscript has several substantive inconsistencies and major problems that need to be addressed prior to a final editorial decision.

Response: We very much appreciate the reviewer's thoughtful and helpful comments. During the past month, we have performed more experiments to acquire more data. All these data are added to the manuscript. Moreover, we have also done a series of modifications/corrections/additions to the manuscript. We wish that this revised version can now address all the concerns raised by the respected reviewer and satisfy the high publication standard in *Nature Communications*. Below please also find our point-by-point responses.

1. The term self-powered catalytic therapy is not suitable as the catalytic activity is mostly triggered by ultrasound irradiation. Hence, this term should be justified in more appropriate way as this is not thermo responsive therapy triggered by the body heat which could be called self-powered.

Response: Thank you for your comments. The term "self-powered" has been deleted in our revised manuscript. And the title of this manuscript has been revised to "**Two-Dimensional Interplanar Heterojunction Mediated Efficient Catalytic Cancer Therapy**"

2. The overall manuscript has no description of piezo catalytic activity, if the activity is US driven, how is it different from piezo catalytic therapy?

3. If the activity is piezo catalytic, which seems it is, the authors should demonstrate the piezo response of the material such as PFM study.

Response: Thank you for your professional comments. Although the piezo catalysis and our developed catalytic therapy are triggered by US, the mechanisms behind them are completely different. In order to make readers better understand, we re-explain the mechanisms of these two catalytic systems in more details. Additionally, PFM study was also carried out to further demonstrate the mechanism of this catalytic reaction.

"One of the most popular US-mediated catalysis is piezo-catalysis relied on piezoelectric materials.³⁹⁻⁴⁰ A piezocatalytic reaction is generally due to piezoelectric polarization generated by piezoelectric materials triggering an electrochemical process. Piezoelectric materials with noncentrosymmetric structures possess electric

dipoles, which will form an external electric field under mechanical deformation, such as US irradiation.⁴¹ Hence, in order to dig the mechanism of FeOCl/FeOOH NSs mediated catalysis under US irradiation, the piezoelectric properties of FeOCl/FeOOH NSs were characterized in details. To probe piezoelectric properties in FeOCl/FeOOH NSs, piezoresponse force microscopy (PFM) technique has been employed. A localized point to point piezo-response of FeOCl/FeOOH NSs has been probed by applying a probe bias of ± 8 V. Supplementary Fig. 11a and 11b represent PFM amplitude and phase signals, respectively. However, amplitude butterfly loop and phase hysteresis, the representative properties of piezoelectric materials, were not observed in Supplementary Fig. 11a and 11b, which illustrated there was no piezoelectric properties in FeOCl/FeOOH NSs and piezo-catalysis might not be the main mechanism behind FeOCl/FeOOH NSs. Another possible mechanism of this reaction is sonodynamic or sonocatalytic effect relied on sonosensitizers or sonocatalysts. The sonosensitizers or sonocatalysts are mainly semiconductors with suitable band structure, which are similar to photosensitizers or photocatalysts. When traveling through a liquid environment or tissue, the US will induce gas-filled microbubbles to oscillate in the acoustic field. This process is called cavitation and with the increase of the acoustic pressure, the microbubbles will finally implode. Upon implosion of the microbubbles, lots of heat and sometimes light (sonoluminescence) is released. Such an energy in a short period of time is able to trigger the separation of electron-hole pairs of sonosensitizers to generate ROS by catalyzing redox reaction with surrounding substrates.”

- “39. Lin, Y. J., Khan, I., Saha, S., Wu, C. C., Barman, S. R., Kao, F. C., Lin, Z. H. Thermocatalytic hydrogen peroxide generation and environmental disinfection by Bi₂Te₃ nanoplates. *Nat. Commun.* **12**, 180 (2021).
40. Lai, Y.-H., Chen, Y.-H., Pal, A., Chou, S.-H., Chang, S.-J., Huang, E. W., Lin, Z.-H., Chen, S.-Y. Regulation of cell differentiation via synergistic self-powered stimulation and degradation behavior of a biodegradable composite piezoelectric scaffold for cartilage tissue. *Nano Energy* **90**, 106545 (2021).
41. Wang, X., Song, J., Liu, J., Wang, Z. L. Direct-Current Nanogenerator Driven by Ultrasonic Waves. *Science* **316**, 102-105 (2007).”

Supplementary Figure 11. a PFM amplitude butterfly loop and **b** PFM phase hysteresis loop of the FeOCl/FeOOH NSs.

4. The manuscript is entirely written highlighting the material point of view and there is negligible amount of importance mentioned from the tumor point of view. The authors should clearly describe the motivation in the story as to why this study is important for treating tumors and why the particular target?

Response: The reviewer's comment is very constructive and useful. The motivation of this story from the tumor point of view has been added in Introduction section.

“High expression of H_2O_2 is one of the unique microenvironments of tumor cells and a specific target of chemodynamic tumor therapy. However, the endogenous H_2O_2 is still not nearly enough to meet the demand for satisfactory chemodynamic efficacy. Moreover, the reported H_2O_2 supplying strategies to improve chemodynamic efficacy were restricted to safety, targeting, and tumor hypoxic microenvironment. Hence, engineering CDT agents with specific and efficient H_2O_2 -supplying functionality without being limited to the TME is a huge breakthrough for clinic development of CDT and cancer treatment. Here, we develop a smart wet-chemical strategy based on an alkali replacement reaction that can intelligently integrate interplanar heterojunction synthesis and two-dimensional ultrathin heterojunction exfoliation in one step.....”

5. The authors also need to mention how and why this reported work is beneficial and advantageous over other catalytic techniques.

Response: Thank you for your comments. The advantages of our reported catalytic therapy over other catalytic therapies have been added in Discussion section.

“Cancer is one of the biggest killers of human health.⁴⁸ Catalysis, a versatile “tool”, has created a colorful world for human beings' survival and development over the past several centuries. In the past decade, cross-amalgamation catalysis and medicine, named catalytic medicine or catalytic therapy, have integrated catalytic technology to solve medical problems, providing effective therapeutic strategies for a variety of diseases, especially cancer.⁴⁹ According to different catalytic mechanisms, catalytic therapy is divided into two categories: exogenous excitation catalysis and nonexogenous excitation catalysis.⁵⁰ Among them, exogenous excitation catalytic therapies mainly refer to the catalytic reactions triggered by specific external stimuli, such as light and US. Photodynamic therapy is the most representative light-triggered catalytic therapy, which severely limited by the poor penetration of light. US has been broadly used in clinic for disease diagnosis and treatment based on its noninvasiveness, minor energy attenuation and high tissue-penetrating capability, such as US imaging, sonodynamic therapy, and piezocatalytic therapy.⁵¹ Piezocatalytic therapy relying on the piezoelectric polarization generated by piezoelectric materials triggering an electrochemical process, holds great potential in tumor therapy and other biomedical fields. However, the high demand for piezoelectric materials has limited the development of piezocatalytic therapy. Sonodynamic or sonocatalytic therapy mainly refers to the electron-hole pair

separation of catalysts under US irradiation, and use the redox capacity of the separated electrons or holes to catalyze a certain redox reaction to produce therapeutic products. The broad spectrum of the catalyst and the deep penetration of US give sonodynamic/sonocatalytic therapy unparalleled advantages.

As one of the most representative nonexogenous excitation catalysis, CDT, applying relatively high expression of H_2O_2 in tumor cells as target and substrate for production of highly cytotoxic $\cdot\text{OH}$, is a burgeoning and promising cancer treatment strategy. Nevertheless, due to the poor catalytic activity of traditional CDT agents and limited substrate concentration, currently CDT alone cannot achieve satisfactory efficacy, and is often used as a complementary therapy combined with other therapies. During past decades, despite enormous struggles, engineering CDT agents with high catalytic activity, specific and efficient H_2O_2 -supplying functionality without being limited to the TME remains a great challenge.

Hence, we presented a novel and collaborative combination, in which 2D interplanar heterojunction FeOCl/FeOOH NSs with high Fenton reaction catalyzing activity and US-triggering H_2O_2 supplying ability were fabricated and created a clever combination between CDT and sonodynamic/sonocatalytic therapy. Firstly, a smart wet-chemical method based on alkali replacement reaction was developed to intelligently integrate interplanar heterojunction synthesis and two-dimensional ultrathin heterojunction exfoliation in one step. After PEGylation, the engineering FeOCl/FeOOH NSs functionalized with a characteristic interplanar heterojunction for efficient H_2O_2 self-supplying applying H_2O as the only substrate was developed and exhibited a highly efficient chemodynamic effect. In the interplanar heterojunction FeOCl/FeOOH NSs, FeOCl part and FeOOH part with different Fermi levels and energy band structures contacting with each other induces charges to redistribute at their interfaces on account of the aligning Fermi levels, which mediates the construction of a built-in electric field at their interface. Under US irradiation, the US-excited electrons on the CB of FeOCl will be combined with the holes on the VB of FeOOH guided by the built-in electric field at their interface, leaving stronger reduction/oxidation potentials of separated electrons and holes on the CB of FeOOH and the VB of FeOCl. Meanwhile, a Schottky barrier was formed owing to band bending at their interface, which will prevent the electron flow from FeOOH to FeOCl and enhanced their Z-schemed charges transfer. Hence, a built-in electric field and Schottky barrier facilitated Z-schemed catalytic mechanism was constructed, in which the holes on the VB of FeOCl have great ability to oxidize H_2O and produce O_2 , meanwhile, the generated O_2 will be immediately and directly reduced to H_2O_2 by the electrons on the CB of FeOOH. The self-supplying H_2O_2 guarantees the continuously and endlessly $\cdot\text{OH}$ generation *via* the disproportionated reaction catalyzed by both FeOCl and FeOOH. The obtained FeOCl/FeOOH NSs with highly efficient chemodynamic effects exhibited excellent anti-cancer performance both *in vitro* and *in vivo*. Therefore, this research not only defines an intelligent strategy for the intelligent synthesis of 2D ultrathin heterojunction, but also gives evidence of the proof-of-concept application of the engineering 2D NSs in biomedical application, which may also provide a certain incentive to make valuable contributions in other

possible fields in the future. Since this 2D interplanar heterojunction uses H₂O as the only needed exogenous substrate and US exposure triggering H₂O₂ and ·OH generation guarantees the targeting and specificity and this catalytic therapy, 2D interplanar heterojunction FeOCl/FeOOH NSs mediated catalytic therapy should possess huge potentials in other biomedical fields, such as sterilization and anti-infection of diabetic wound, operative wound, and other infections.”

- “48. Sung, H., Ferlay, J., Siegel, R. L., Laversanne, M., Soerjomataram, I., Jemal, A., Bray, F. Global cancer statistics 2020: GLOBOCAN estimates of incidence and mortality worldwide for 36 cancers in 185 countries. *Ca- Cancer J. Clin.* **71**, 209-249 (2021).
49. Yang, B., Chen, Y., Shi, J. Nanocatalytic Medicine. *Adv. Mater.* **31**, 1901778 (2019).
50. Pan, C., Mao, Z., Yuan, X., Zhang, H., Mei, L., Ji, X. Heterojunction Nanomedicine. *Adv. Sci.* 2105747 (2022).
51. Zhu, P., Chen, Y., Shi, J. Piezocatalytic Tumor Therapy by Ultrasound-Triggered and BaTiO₃ -Mediated Piezoelectricity. *Adv. Mater.* **32**, 2001976 (2020).”

6. The captions in the figure 2 are repetitive and very confusing for example f,g, and h are captioned twice and it is confusing in the results and discussion section what the authors are trying to convey. These figures should be re captioned and the discussion should be re-written accordingly.

Response: Thanks for this valuable comment. We are very sorry for this negligence. The typographical errors in Fig. 2 and the corresponding discussion have been revised.

Fig. 2 Morphology and composition characterization of layered FeOCl NSs and FeOCl/FeOOH NSs. SEM images of layered **a** FeOCl and **e** FeOCl/FeOOH, scale bar = 100 nm. Enlarged SEM images of layered **b** FeOCl and **f** FeOCl/FeOOH, scale bar = 100 nm. TEM images of ultrathin **c** FeOCl NSs and **g** FeOCl/FeOOH NSs, scale bar = 15 nm. HRTEM images of ultrathin **d** FeOCl NSs and **h** FeOCl/FeOOH NSs, scale bar = 3 nm. SEM-EDS mapping of **i** FeOCl and **j** FeOCl/FeOOH: red (Fe), green (O), and yellow (Cl), scale bar = 100 nm. **k** AFM images of FeOCl/FeOOH NSs, scale bar = 100 nm. **l** Thickness of FeOCl/FeOOH NSs. **m** 3D AFM images of FeOCl/FeOOH NSs, scale bar = 100 nm.

7. In figure 3, XRD indexing is missing and even in the description section, the phases are not mentioned which is not clear for readers. Also, the same for Raman spectroscopy this should be mentioned and explained in detail.

Response: We appreciate this helpful comment. These important detailed analyses were added in our revised manuscript.

The characterizations of FeOCl/FeOOH heterojunction including XRD and Raman analysis:

“In the XRD spectra of FeOCl NSs (Fig. 3b), all the XRD peaks are well-matched with JCPDS card No. 01-0081 corresponding to orthorhombic structured FeOCl nanocrystals, which demonstrated the high purity of synthesized FeOCl. After alkali etching, another respective crystal structures were observed, which corresponded with FeOOH (JCPDS No. 01-0136), illustrating the successfully edge decoration. In the Raman spectra (Fig. 3c), the peaks of FeOCl at A_{1g} mode (212 cm^{-1}) and E_g mode (291 cm^{-1}) are related to Fe–O stretching vibration. After alkali etching, the characteristic peaks of FeOOH at 240 cm^{-1} and 400 cm^{-1} were all exhibited in the spectrum of FeOCl/FeOOH NSs, which further demonstrated the successfully edge decoration.”

Fig. 3 Chemical composition and structure characterization of layered FeOCl NSs and FeOCl/FeOOH NSs. a XPS spectra of FeOCl NSs and FeOCl/FeOOH NSs. **b** XRD spectra of FeOCl NSs and FeOCl/FeOOH NSs. **c** Raman shift spectra of FeOCl NSs and FeOCl/FeOOH NSs. **d** and **g** HRXPS spectra of Fe 2p in FeOCl NSs and FeOCl/FeOOH NSs. **e** and **h** HRXPS spectra of Cl 2p in FeOCl NSs and FeOCl/FeOOH NSs. **f** and **i** HRXPS spectra of O 1s in FeOCl NSs and FeOCl/FeOOH NSs.

8. Since PEGylation of the NSs is done for the final study, all the materials characterizations should be repeated after PEGylation and compared to non PEGylation samples as to prove that the materials property has not been affected. This is an important point, which the authors must address.

9. Supplementary figure 4 should be clearly captioned and addressed in the manuscript clearly as well.

Response: Thank you for your comments. We are sorry for this misunderstanding. Actually, all the materials characterizations, including catalytic performance *in vitro*, all cell experiments, and all *in vivo* experiments, were carried out after PEGylation. The reason for the PEGylation of nanomedicines and the influence of PEGylation for nanomedicines' catalytic efficiency and antitumor performance have been added in our revised manuscript. In addition, the caption of supplementary Fig. 4 and the corresponding discussion have been revised.

“Due to the biocompatibility and dispersibility are crucial for biomedical applications, the as-prepared FeOCl NSs and FeOCl/FeOOH NSs were modified by PEG-NH₂. PEG(5k)-NH₂ positively charged was absorbed on the negatively charged surface of prepared FeOCl NSs and FeOCl/FeOOH NSs (Supplementary Fig. 3) *via* electrostatic attraction. The UV-vis-NIR absorbance spectra exhibited that there was no significant difference before and after PEG modification (Supplementary Fig. 4). The amount of PEG(5k)-NH₂ that was coated on the surface of the FeOCl/FeOOH NSs was $\approx 22.4\%$ (w/w) as measured by thermo gravimetric analysis (TGA). PEGylation of FeOCl NSs and FeOCl/FeOOH NSs possess an enhanced dispersion in water, phosphate buffer saline (PBS), and cell culture medium compared with bare NSs (Supplementary Fig. 5). So that, more catalytic active sites of PEGylated nanocatalyst were exposed to substrates than that of aggregated nanocatalysts. In addition, since the catalytic substrates are dissolved in water, PEG modification can effectively increase the surface hydrophilicity of nanomaterials, and thus increase the adsorption of substrate molecules, such as dissolved O₂, GSH, and H₂O, on the surface of prepared NSs. Therefore, PEGylation of prepared NPs or other nanocatalysts is necessary for safeguarding their catalytic activity. More important, for intravenously injected nanomedicines, PEG modification is an important means to ensure that nanomedicines are not cleared by immune cells, so as to fully guarantee the circulation time and tumor enrichment of nanomedicines *in vivo*. Therefore, PEGylation is essential for nanomedicine used *in vivo*, and all of the catalytic properties, *in vitro*, and *in vivo* experiments in this research were carried out after PEGylation of FeOCl NSs and FeOCl/FeOOH NSs.”

Supplementary Figure 3. The Zeta potentials of prepared FeOCl NSs and FeOCl/FeOOH NSs.

Supplementary Figure 5. The distribution of **a** FeOCl NSs, **b** PEGylated FeOCl NSs, **c** FeOCl/FeOOH NSs, and **d** PEGylated FeOCl/FeOOH NSs in (1) water, (2) PBS, (3) fetal bovine serum solution, and (4) medium.

10. The authors have focused on the self-supplying H_2O_2 ability of the material. However, the authors need to provide the H_2O_2 as well as $\cdot\text{OH}$ production data of their material in water with and without ultrasound. The data should be provided with and without PEGylation of the materials. Authors may refer to (Nature Communications volume 12, Article number: 180 (2021) and Nano Energy 57 (2019) 14–2115 for these experiments.

Response: Thank you for this suggestion. Referring to these two previous reports, the H_2O_2 and $\cdot\text{OH}$ production data catalyzed by FeOCl NSs and FeOCl/FeOOH NSs with or without US irradiation and with or without PEGylation have been detected and provided in our revised manuscript.

Supplementary Figure 8. The **a** $\cdot\text{OH}$ and **b** H_2O_2 production data catalyzed by FeOCl NSs and FeOCl/FeOOH NSs with or without US irradiation and with or without PEGylation.

“Moreover, the H_2O_2 and $\cdot\text{OH}$ production data catalyzed by FeOCl NSs and FeOCl/FeOOH NSs with or without US irradiation and with or without PEGylation were also detected and quantitative analysis. As shown in Supplementary Fig. 8a and 8b, the quantitative H_2O_2 and $\cdot\text{OH}$ production data were identified with the data exhibited in Fig. 4a and 4c. Additionally, under the same condition, FeOCl NSs and FeOCl/FeOOH NSs with PEGylation displayed higher catalytic activity compared to that without PEGylation. The enhanced dispersibility endowed by PEGylation providing much more catalytic active sites should be the main reason.”

11. In figure 4c, d no generation of in H_2O_2 FeOCl is due to unfavorable band position of FeOCl to generate H_2O_2 as shown in figure 5g. Authors should rectify this description in terms of band position.

Response: Thank you for your comments. The description has been revised accordingly.

“For FeOCl NSs + US group without extra added H₂O₂, there was also no H₂O₂ generation, demonstrating the FeOCl NSs exposed to the US cannot generate H₂O₂ due to unfavorable band position of FeOCl to generate H₂O₂.”

12. The position for ·OH production described in the schematic of figure 5g is incorrect and should be re-checked.

Response: Thank you for your valuable comments. The mechanism and position for ·OH production in the schematic of Fig. 5g has been revised.

Fig. 5 Catalytic mechanism analysis of FeOCl NSs and FeOCl/FeOOH NSs. a UV-Vis absorbance spectra and **b** calculated band gap of FeOCl NSs and FeOCl/FeOOH NSs. **c** Valence band of FeOCl NSs and FeOCl/FeOOH NSs calculated from XPS spectra. Density functional theory (DFT) computational calculations of **d** FeOCl and **e** FeOOH. **f** Schematic illustration and chemical reaction equations of Z-schemed heterojunction based on FeOCl/FeOOH NSs for ·OH generation using water as the only substrate. **g** Mechanism of Z-schemed heterojunction based on FeOCl/FeOOH NSs for ·OH generation.

13. The authors should also check if the hydroxyl radical generation is the result of band tailoring or fenton reactions using the generated H₂O₂. Authors may confirm this by exactly evaluating the favorable bond position for the ROS generation.

Response: We very much appreciate the reviewer's valuable comments. The related discuss has been added in our revised manuscript.

“To understand the microscopic mechanism of the hydroxyl radical generation, we calculated the Gibbs free energy change (ΔG) of the H₂O pathway and the H₂O₂ pathway, respectively, using DFT method. The ΔG is an effective parameter to character the catalytic activity, which follows the expression below,

$$\Delta G = \Delta E + \Delta E_{ZPE} - T\Delta S \quad (1)$$

where ΔE , ΔE_{ZPE} , T, and ΔS stand for the electronic energy difference, the zero-point energy difference, the human body temperature (310.15 K) and the entropy difference, respectively. The zero-point energy was obtained by the vibration frequency calculations. In a vibration frequency model, adsorbate species is released and the substrate is fixed due to the insignificant vibration of the substrate.

Two possible reaction pathways for the hydroxyl radical generation starting from H₂O and H₂O₂, respectively, are listed below,

where * represents an adsorption site on the catalyst which is FeOCl phase in H₂O pathway and FeOOH phase in H₂O₂ pathway, *OH represents the adsorbed OH on the catalytic site. In step (a), the free energy of the electron-proton pair was considered equal to that of 1/2 H₂ at ambient conditions. In step (c), Fenton reaction product, Fe(III)-hydroxide pair was considered equal to *OH.

The schematic diagram of catalytic pathways for ·OH generation is illustrated in Supplementary Fig. 9. For the H₂O pathway, one H₂O molecule undergoes steps (a) and (c) successively to generate one ·OH molecule. The ΔG of step (a) and (c) are 3.14 eV and -0.27 eV, respectively, meaning that the rate-limiting step to generate ·OH from H₂O is step (a) with an energy barrier of 3.14 eV. For H₂O₂ pathway, one H₂O₂ molecule generate one ·OH molecule via step (c) with an energy barrier of 1.57 eV. Although the two proposed catalytic reaction pathways are endothermic processes, the H₂O₂ pathway has a much lower energy barrier and less energy requirement, so a more favourable ·OH generation mechanism is that the H₂O₂ molecules are catalyzed by FeOOH via one-step Fenton reaction.”

Supplementary Figure 9. Schematic diagram of catalytic pathways for $\cdot\text{OH}$ generation. The free energy difference of each elementary reaction is shown, and the energy of the initial component is set to be 0 eV.

14. For in vitro experiments, amount of H_2O_2 used was significantly high and for in vivo experiments no externally supplied H_2O_2 was used. The amount of H_2O_2 in the TME may not be sufficiently high and in this case, what will be the anti-tumor performance of the material. Was H_2O_2 also supplied along with the material in the in vivo experiments?

Response: Thank you for your comments. For the in vitro experiments, additional H_2O_2 was added only for FeOCl + US group. Externally added H_2O_2 in FeOCl + US group was employed to prove the internal producing H_2O_2 ability of FeOCl/FeOOH NSs under US irradiation (FeOCl/FeOOH + US group). Due to the internal in situ H_2O_2 -supplying ability of FeOCl/FeOOH NSs under US irradiation, a satisfactory antitumor effect was achieved for FeOCl/FeOOH + US group without externally added H_2O_2 .

“Interestingly, Z-scheme FeOCl/FeOOH NSs based interplanar heterojunction exposed to US irradiation exhibited the highest cytotoxicity to cancer cells, in which more than 90% tumor cells were dead with 100 $\mu\text{g}/\text{mL}$ FeOCl/FeOOH NSs and 10 mins’ US irradiation. The main cause of this excellent anti-tumor performance of FeOCl/FeOOH NSs should attribute to the supplying H_2O_2 through two steps cascade reaction ($\text{H}_2\text{O} \xrightarrow{h^+} \text{O}_2 \xrightarrow{e^-} \text{H}_2\text{O}_2$). To further test the influence of H_2O_2 concentration

on the chemodynamic effect, a certain amount of H_2O_2 was slowly added to maintain the H_2O_2 concentration in TME in FeOCl NSs + US group. As shown in Fig. 6b, similar cytotoxicity of FeOCl NSs + US group was observed compared with FeOCl/FeOOH NSs + US group, which not only demonstrates the H_2O_2 concentration

is vital for chemodynamic effect, but also confirm the excellent H₂O₂ supplying ability of FeOCl/FeOOH NSs.”

15. In most of the experiments H₂O₂ was externally supplied, thus H₂O as the only substrate may not be a reasonable claim. Authors should rethink about this claim with proper justification.

Response: Thank you for your comments. As explained to question 14, for catalytic characterizations and *in vitro* antitumor experiments, additional H₂O₂ was added only for FeOCl+US group. Externally added H₂O₂ in FeOCl + US group was employed to prove the internal producing H₂O₂ ability of FeOCl/FeOOH NSs under US irradiation (FeOCl/FeOOH + US group). In this research, we constructed 2D interplanar heterojunction FeOCl/FeOOH NSs, which were able to producing H₂O₂ under US irradiation through two steps cascade reaction ($\text{H}_2\text{O} \xrightarrow{h^+} \text{O}_2 \xrightarrow{e^-} \text{H}_2\text{O}_2$) using H₂O as the only substrate. Moreover, the highest catalytic efficiency of ·OH generation and the best tumor inhibition effect *in vitro* and *in vivo* were achieved applying FeOCl/FeOOH + US treatment.

16. It will be an interesting study as how this therapy can be applied on diabetic subjects. What according to the authors will be a challenge or will the catalytic performance remain unchanged irrespective of the subject’s health conditions?

Response: Thank you for this constructive comment. Except cancer treatment, this catalytic therapy possesses huge potentials in other biomedical fields, such as sterilization and anti-infection of diabetic wound, operative wound, and other infections. The corresponding statements have been added in our revised manuscript. “Therefore, this research not only defines an intelligent strategy for the intelligent synthesis of 2D ultrathin heterojunction, but also gives evidence of the proof-of-concept application of the engineering 2D NSs in biomedical application, which may also provide a certain incentive to make valuable contributions in other possible fields in the future. Since this 2D interplanar heterojunction uses H₂O as the only needed exogenous substrate and US exposure triggering H₂O₂ and ·OH generation guarantees the targeting and specificity and this catalytic therapy, 2D interplanar heterojunction FeOCl/FeOOH NSs mediated catalytic therapy should possess huge potentials in other biomedical fields, such as sterilization and anti-infection of diabetic wound, operative wound, and other infections.”

Reviewers' Comments:

Reviewer #1:

Remarks to the Author:

The authors have addressed all my concerns. It can be accepted now.

Reviewer #2:

Remarks to the Author:

The authors have made revisions accordingly. With these improvements and clarification I can now recommend for publication.

Reviewer #3:

Remarks to the Author:

The authors have clearly responded to my comments and made good revisions according to my suggestions. Therefore, I would like to suggest the acceptance of this manuscript.

Reviewers' comments:

REVIEWERS' COMMENTS:

Reviewer #1 (Remarks to the Author):

The authors have addressed all my concerns. It can be accepted now.

Response: Thank you very much for your kind words after revision.

Reviewer #2 (Remarks to the Author):

The authors have made revisions accordingly. With these improvements and clarification I can now recommend for publication.

Response: Thank you very much for your nice comment.

Reviewer #3 (Remarks to the Author):

The authors have clearly responded my comments and made good revisions according to my suggestions. Therefore, I would like to suggest the acceptance of this manuscript.

Response: Thank you very much for your nice comment.